# CAPACITY OF GROUP-INVARIANT LINEAR READOUTS FROM EQUIVARIANT REPRESENTATIONS: HOW MANY OBJECTS CAN BE LINEARLY CLASSIFIED UNDER ALL POSSIBLE VIEWS?

**Matthew Farrell**[*‡], **Blake Bordelon**[*‡], **Shubhendu Trivedi**[†], **& Cengiz Pehlevan**[‡]

[‡] Harvard University
{msfarrell,blake_bordelon,cpehlevan}@seas.harvard.edu

[†] Massachusetts Institute of Technology
shubhendu@csail.mit.edu

## ABSTRACT

Equivariance has emerged as a desirable property of representations of objects subject to identity-preserving transformations that constitute a group, such as translations and rotations. However, the expressivity of a representation constrained by group equivariance is still not fully understood. We address this gap by providing a generalization of Cover's Function Counting Theorem that quantifies the number of linearly separable and group-invariant binary dichotomies that can be assigned to equivariant representations of objects. We find that the fraction of separable dichotomies is determined by the dimension of the space that is fixed by the group action. We show how this relation extends to operations such as convolutions, element-wise nonlinearities, and global and local pooling. While other operations do not change the fraction of separable dichotomies, local pooling decreases the fraction, despite being a highly nonlinear operation. Finally, we test our theory on intermediate representations of randomly initialized and fully trained convolutional neural networks and find perfect agreement.

## 1 INTRODUCTION

The ability to robustly categorize objects under conditions and transformations that preserve the object categories is essential to animal intelligence, and to pursuits of practical importance such as improving computer vision systems. However, for general-purpose understanding and geometric reasoning, invariant representations of these objects in sensory processing circuits are not enough. Perceptual representations must also accurately encode their transformation properties.

One such property is that of exhibiting equivariance to transformations of the object. When such transformations are restricted to be an algebraic group, the resulting equivariant representations have found significant success in machine learning starting with classical convolutional neural networks (CNNs) (Denker et al., 1989; LeCun et al., 1989) and recently being generalized by the influential work of Cohen & Welling (2016). Such representations have elicited burgeoning interest as they capture many transformations of practical interest such as translations, permutations, rotations, and reflections. Furthermore, equivariance to these transformations can be easily "hard-coded" into neural networks. Indeed, a new breed of CNN architectures that explicitly account for such transformations are seeing diverse and rapidly growing applications (Townshend et al., 2021; Baek et al., 2021; Satorras et al., 2021; Anderson et al., 2019; Bogatskiy et al., 2020; Klicpera et al., 2020; Winkels & Cohen, 2019; Gordon et al., 2020; Sosnovik et al., 2021; Eismann et al., 2020). In addition, equivariant CNNs have been shown to capture response properties of neurons in the primary visual cortex beyond classical Gábor filter models (Ecker et al., 2018).

---

[*]These authors contributed equally.

While it is clear that equivariance imposes a strong constraint on the geometry of representations and thus of perceptual manifolds (Seung & Lee, 2000; DiCarlo & Cox, 2007) that are carved out by these representations as the objects transform, the implications of such constraints on their expressivity are not well understood. In this work we take a step towards addressing this gap. Our starting point is the classical notion of the **perceptron capacity** (sometimes also known as the fractional memory/storage capacity) – a quantity fundamental to the task of object categorization and closely related to VC dimension (Vapnik & Chervonenkis, 1968). Defined as the maximum number of points for which all (or 1-$\delta$ fraction of) possible binary label assignments (i.e. **dichotomies**) afford a hyperplane that separates points with one label from the points with the other, it can be seen to offer a quantification of the expressivity of a representation.

Classical work on perceptron capacity focused on points in general position (Wendel, 1962; Cover, 1965; Schläfli, 1950; Gardner, 1987; 1988). However, understanding the perceptron capacity when the inputs are not merely points, but are endowed with richer structure, has only recently started to attract attention. For instance, work by Chung et al. (2018); Pastore et al. (2020); Rotondo et al. (2020); Cohen et al. (2020) considered general perceptual manifolds and examined the role of their geometry to obtain extensions to the perceptron capacity results. However, such work crucially relied on the assumption that each manifold is oriented randomly, a condition which is strongly violated by equivariant representations.

With these motivations, our particular contributions in this paper are the following:

- We extend Cover's function counting theorem and VC dimension to equivariant representations, finding that both scale with the dimension of the subspace fixed by the group action.
- We demonstrate the applicability of our result to $G$-convolutional network layers, including pooling layers, through theory and verify through simulation.

## 1.1 RELATED WORKS

Work most related to ours falls along two major axes. The first follows the classical perceptron capacity result on the linear separability of points (Schläfli, 1950; Wendel, 1962; Cover, 1965; Gardner, 1987; 1988). This result initiated a long history of investigation in theoretical neuroscience, e.g. (Brunel et al., 2004; Chapeton et al., 2012; Rigotti et al., 2013; Brunel, 2016; Rubin et al., 2017; Pehlevan & Sengupta, 2017), where it is used to understand the memory capacity of neuronal architectures. Similarly, in machine learning, the perceptron capacity and its variants, including notions for multilayer perceptrons, have been fundamental to a fruitful line of study in the context of finite sample expressivity and generalization (Baum, 1988; Kowalczyk, 1997; Sontag, 1997; Huang, 2003; Yun et al., 2019; Vershynin, 2020). Work closest in spirit to ours comes from theoretical neuroscience and statistical physics (Chung et al., 2018; Pastore et al., 2020; Rotondo et al., 2020; Cohen et al., 2020), which considered general perceptual manifolds, albeit oriented randomly, and examined the role of their geometry to obtain extensions to the perceptron capacity result.

The second line of relevant literature is that on group-equivariant convolutional neural networks (GCNNs). The main inspiration for such networks comes from the spectacular success of classical CNNs (LeCun et al., 1989) which directly built in translational symmetry into the network architecture. In particular, the internal representations of a CNN are approximately[1] translation equivariant: if the input image is translated by an amount $t$, the feature map of each internal layer is translated by the same amount. Furthermore, an invariant read-out on top ensures that a CNN is translation invariant. Cohen & Welling (2016) observed that a viable approach to generalize CNNs to other data types could involve considering equivariance to more general transformation groups. This idea has been used to construct networks equivariant to a wide variety of transformations such as planar rotations (Worrall et al., 2017; Weiler et al., 2018b; Bekkers et al., 2018; Veeling et al., 2018; Smets et al., 2020), 3D rotations (Cohen et al., 2018; Esteves et al., 2018; Worrall & Brostow, 2018; Weiler et al., 2018a; Kondor et al., 2018a; Perraudin et al., 2019), permutations (Zaheer et al., 2017; Hartford et al., 2018; Kondor et al., 2018b; Maron et al., 2019a; 2020), general Euclidean isometries (Weiler et al., 2018a; Weiler & Cesa, 2019; Finzi et al., 2020), scaling (Marcos et al., 2018; Worrall & Welling, 2019; Sosnovik et al., 2020) and more exotic symmetries (Bogatskiy et al., 2020; Shutty & Wierzynski, 2020; Finzi et al., 2021) etc.

---

[1]Some operations such as max pooling and boundary effects of the convolutions technically break strict equivariance, as well as the final densely connected layers.

A quite general theory of equivariant/invariant networks has also emerged. Kondor & Trivedi (2018) gave a complete description of GCNNs for scalar fields on homogeneous spaces of compact groups. This was generalized further to cover the steerable case in (Cohen et al., 2019b) and to general gauge fields in (Cohen et al., 2019a; Weiler et al., 2021). This theory also includes universal approximation results (Yarotsky, 2018; Keriven & Peyré, 2019; Sannai et al., 2019b; Maron et al., 2019b; Segol & Lipman, 2020; Ravanbakhsh, 2020). Nevertheless, while benefits of equivariance/invariance in terms of improved sample complexity and ease of training are quoted frequently, a firm theoretical understanding is still largely missing. Some results however do exist, going back to (Shawe-Taylor, 1991). Abu-Mostafa (1993) made the argument that restricting a classifier to be invariant can not increase its VC dimension. Sokolic et al. (2017) extend this idea to derive generalization bounds for invariant classifiers, while Sannai et al. (2019a) do so specifically working with the permutation group. Elesedy & Zaidi (2021) show a strict generalization benefit for equivariant linear models, showing that the generalization gap between a least squares model and its equivariant version depends on the dimension of the space of anti-symmetric linear maps. Some benefits of related ideas such as data augmentation and invariant averaging are formally shown in (Lyle et al., 2020; Chen et al., 2020). Here we focus on the limits to expressivity enforced by equivariance.

## 2 PROBLEM FORMULATION

Suppose $x$ abstractly represents an object and let $r(x) \in \mathbb{R}^N$ be some feature map of $x$ to an $N$-dimensional space (such as an intermediate layer of a deep neural network). We consider transformations of this object, such that they form a group in the algebraic sense of the word. We denote the abstract transformation of $x$ by element $g \in G$ as $gx$. Groups $G$ may be represented by invertible matrices, which act on a vector space $V$ (which themselves form the group $GL(V)$ of invertible linear transformations on $V$). We are interested in feature maps $r$ which satisfy the following group equivariance condition:

$$r(gx) = \pi(g)r(x),$$

where $\pi : G \to GL(\mathbb{R}^N)$ is a linear **representation** of $G$ which acts on feature map $r(x)$. Note that many representations of $G$ are possible, including the trivial representation: $\pi(g) = I$ for all $g$.

We are interested in perceptual object manifolds generated by the actions of $G$. Each of the $P$ manifolds can be written as a set of points $\{\pi(g)r^\mu : g \in G\}$ where $\mu \in [P] \equiv \{1, 2, \ldots, P\}$; that is, these manifolds are orbits of the point $r^\mu \equiv r(x^\mu)$ under the action of $\pi$. We will refer to such manifolds as $\pi$**-manifolds**.[2]

Each of these $\pi$-manifolds represents a single object under the transformation encoded by $\pi$; hence, each of the points in a $\pi$-manifold is assigned the same class label. A perceptron endowed with a set of linear readout weights $w$ will attempt to determine the correct class of every point in every manifold. The condition for realizing (i.e. linearly separating) the dichotomy $\{y^\mu\}_\mu$ can be written as $y^\mu w^\top \pi(g)r^\mu > 0$ for all $g \in G$ and $\mu \in [P]$, where $y^\mu = +1$ if the $\mu^{\text{th}}$ manifold belongs to the first class and $y^\mu = -1$ if the $\mu^{\text{th}}$ manifold belongs to the second class. The perceptron capacity is the fraction of dichotomies that can be linearly separated; that is, separated by a hyperplane.

For concreteness, one might imagine that each of the $r^\mu$ is the neural representation for an image of a dog (if $y^\mu = +1$) or of a cat (if $y^\mu = -1$). The action $\pi(g)$ could, for instance, correspond to the image shifting to the left or right, where the size of the shift is given by $g$. Different representations of even the same group can have different coding properties, an important point for investigating biological circuits and one that we leverage to construct a new GCNN architecture in Section 5.

## 3 PERCEPTRON CAPACITY OF GROUP-GENERATED MANIFOLDS

Here we first state and prove our results in the general case of representations of compact groups over $\mathbb{R}^N$. The reader is encouraged to read the proofs, as they are relatively simple and provide valuable intuition. We consider applications to specific group representations and GCNNs in Sections 4 and 5 that follow.

---

[2]Note that for a finite group $G$, each manifold will consist of a finite set of points. This violates the technical mathematical definition of "manifold", but we abuse the definition here for the sake of consistency with related work (Chung et al., 2018; Cohen et al., 2019b); to be more mathematically precise, one could instead refer to these "manifolds" as $\pi$-orbits.

### 3.1 SEPARABILITY OF $\pi$-MANIFOLDS

We begin with a lemma which states that classifying the $P$ $\pi$-manifolds can be reduced to the problem of classifying their $P$ centroids. For the rest of Section 3, we let $\pi : G \to \mathrm{GL}(\mathbb{R}^N)$ be an arbitrary linear representation of a compact group $G$.[3] We denote the average of $\pi$ over $G$ with respect to the Haar measure by $\langle \pi(g) \rangle_{g \in G}$; for finite $G$ this is simply $\frac{1}{|G|} \sum_{g \in G} \pi(g)$ where $|G|$ is the order (i.e. number of elements) of $G$. For ease of notation we will generally write $\langle \pi \rangle \equiv \langle \pi(g) \rangle_{g \in G}$ when the group $G$ being averaged over is clear.

**Lemma 1.** *A dataset $\{(\pi(g)\boldsymbol{r}^\mu, y^\mu)\}_{g \in G, \mu \in [P]}$ consisting of $P$ $\pi$-manifolds with labels $y^\mu$ is linearly separable if and only if the dataset $\{(\langle \pi \rangle \boldsymbol{r}^\mu, y^\mu)\}_{\mu \in [P]}$ consisting of the $P$ centroids $\langle \pi \rangle \boldsymbol{r}^\mu$ with the same labels is linearly separable. Formally,*

$$\exists \boldsymbol{w} \; \forall g \in G, \mu \in [P] : y^\mu \boldsymbol{w}^\top \pi(g) \boldsymbol{r}^\mu > 0 \iff \exists \boldsymbol{w} \; \forall \mu \in [P] : y^\mu \boldsymbol{w}^\top \langle \pi \rangle \boldsymbol{r}^\mu > 0.$$

*Proof.* The forward implication is obvious: if there exists a $\boldsymbol{w}$ which linearly separates the $P$ manifolds according to an assignment of labels $y^\mu$, that same $\boldsymbol{w}$ must necessarily separate the centroids of these manifolds. This can be seen by averaging each of the quantities $y^\mu \boldsymbol{w}^\top \pi(g) \boldsymbol{r}^\mu$ over $g \in G$. Since each of these quantities is positive, the average must be positive.

For the reverse implication, suppose $y^\mu \boldsymbol{w}^\top \langle \pi \rangle \boldsymbol{r}^\mu > 0$, and define $\tilde{\boldsymbol{w}} = \langle \pi \rangle^\top \boldsymbol{w}$. We will show that $\tilde{\boldsymbol{w}}$ separates the $P$ $\pi$-manifolds since

$$
\begin{aligned}
y^\mu \tilde{\boldsymbol{w}}^\top \pi(g) \boldsymbol{r}^\mu &= y^\mu \boldsymbol{w}^\top \langle \pi \rangle \pi(g) \boldsymbol{r}^\mu \quad \text{(Definition of } \tilde{\boldsymbol{w}}\text{)} \\
&= y^\mu \boldsymbol{w}^\top \langle \pi(g')\pi(g) \rangle_{g' \in G} \boldsymbol{r}^\mu \quad \text{(Definition of } \langle \pi \rangle \text{ and linearity of } \pi(g)\text{)} \\
&= y^\mu \boldsymbol{w}^\top \langle \pi \rangle \boldsymbol{r}^\mu \quad \text{(Invariance of the Haar Measure } \mu(Sg) = \mu(S) \text{ for set } S\text{)} \\
&> 0 \quad \text{(Assumption that } \boldsymbol{w} \text{ separates centroids)}
\end{aligned}
$$

Thus, all that is required to show that $\tilde{\boldsymbol{w}}$ separates the $\pi$-orbits are basic properties of a group representation and invariance of the Haar measure to $G$-transformations. $\qquad\square$

### 3.2 RELATIONSHIP WITH COVER'S THEOREM AND VC DIMENSION

The fraction $f$ of linearly separable dichotomies on a dataset of size $P$ for datapoints in general position [4] in $N$ dimensions was computed by Cover (1965) and takes the form

$$f(P, N) = 2^{1-P} \sum_{k=0}^{N-1} \binom{P-1}{k} \tag{1}$$

where we take $\binom{n}{m} = 0$ for $m > n$. Taking $\alpha = P/N$, $f$ is a nonincreasing sigmoidal function of $\alpha$ that takes the value 1 when $\alpha \leq 1$, 1/2 when $\alpha = 2$, and approaches 0 as $\alpha \to \infty$. In the thermodynamic limit of $P, N \to \infty$ and $\alpha = O(1)$, $f(\alpha)$ converges to a step function $\bar{f}(\alpha) = \lim_{P,N \to \infty, \alpha = P/N} f(P, N) = \Theta(2 - \alpha)$ (Gardner, 1987; 1988; Shcherbina & Tirozzi, 2003), indicating that in this limit all dichotomies are realized if $\alpha < 2$ and no dichotomies can be realized if $\alpha > 2$, making $\alpha_c = 2$ a critical point in this limit.

The VC dimension (Vapnik & Chervonenkis, 1968) of the perceptron trained on points in dimension $N$ is defined to be the largest number of points $P$ such that all dichotomies are linearly separable for some choice of the $P$ points. These points can be taken to be in general position.[5] Because $f(P, N) = 1$ precisely when $P \leq N$, the VC dimension is always $N = P$ for any finite $N$ and $P$ (see Abu-Mostafa et al. (2012)), even while the asymptotics of $f$ reveal that $f(P, N)$ approaches 1 at any $N > P/2$ as $N$ becomes large. In this sense the perceptron capacity yields strictly more information than the VC dimension, and becomes comparatively more descriptive of the expressivity as $N$ becomes large.

In our $G$-invariant storage problem, the relevant scale is $\alpha = P/N_0$ where $N_0$ is the dimension of the fixed point subspace (defined below). We are now ready to state and prove our main theorem.

---

[3] Note that our results extend to more general vector spaces than $\mathbb{R}^N$, under the condition that the group be semi-simple.

[4] A set of $P$ points is in **general position** in $N$-space if every subset of $N$ or fewer points is linearly independent. This says that the points are "generic" in the sense that there aren't any prescribed special linear relationships between them beyond lying in an $N$-dimensional space. Points drawn from a Gaussian distribution with full-rank covariance are in general position with probability one.

[5] This is because linear dependencies decrease the number of separable dichotomies (see Hertz et al. (2018)).

**Theorem 1.** *Suppose the points $\langle\pi\rangle r^\mu$ for $\mu \in [P]$ lie in general position in the subspace $V_0 = \text{range}(\langle\pi\rangle) = \{\langle\pi\rangle x : x \in V\}$. Then $V_0$ is the fixed point subspace of $\pi$, and the fraction of linearly separable dichotomies on the $P$ $\pi$-manifolds $\{\pi(g)r^\mu : g \in G\}$ is $f(P, N_0)$, where $N_0 = \dim V_0$. Equivalently, $N_0$ is the number of trivial irreducible representations that appear in the decomposition of $\pi$ into irreducible representations.*

The **fixed point subspace** is the subspace $W = \{w \in V | gw = w, \forall g \in G\}$. The theorem and its proof use the notion of **irreducible** representations (irreps), which are in essence the fundamental building blocks for general representations. Concretely, any representation of a compact group over a real vector space decomposes into a direct sum of irreps (Naimark & Stern, 1982; Bump, 2004). A representation $\pi : G \to \text{GL}(V)$ is irreducible if $V$ is not 0 and if no vector subspace of $V$ is stable under $G$, other than 0 and $V$ (which are always stable under $G$). A subspace $W$ being stable (or invariant) under $G$ means that $\pi(g)w \in W$ for all $w \in W$ and $g \in G$. The condition that the points $\langle\pi\rangle r^\mu$ be in general position essentially means that there is no prescribed special relationship between the $r^\mu$ and between the $r^\mu$ and $\langle\pi\rangle$. Taking the $r^\mu$ to be drawn from a full-rank Gaussian distribution is sufficient to satisfy this condition.

*Proof.* By the theorem of complete reducibility (see Fulton & Harris (2004)), $\pi$ admits a decomposition into a direct sum of irreps $\pi \cong \pi_{k_1} \oplus \pi_{k_2} \oplus ... \oplus \pi_{k_M}$ acting on vector space $V = V_1 \oplus V_2 \oplus ... V_M$, where $\cong$ denotes equality up to similarity transformation. The indices $k_j$ indicate the type of irrep corresponding to invariant subspace $V_j$. The fixed point subspace $V_0$ is the direct sum of subspaces where trivial $k_j = 0$ irreps act: $V_0 = \bigoplus_{n:k_n=0} V_n$. By the Grand Orthogonality Theorem of irreps (see Liboff (2004)) all non-trivial irreps average to zero $\langle\pi_{k,ij}(g)\rangle_{g \in G} \propto \delta_{k,0}\delta_{i,j}$. Then, the matrix $\langle\pi\rangle$ simply projects the data to $V_0$. By Lemma 1 the fraction of separable dichotomies on the $\pi$-manifolds is the same as that of their centroids $\langle\pi\rangle r^\mu$. Since, by assumption, the $P$ points $\langle\pi\rangle r^\mu$ lie in general position in $V_0$, the fraction of separable dichotomies is $f(P, \dim V_0)$ by Equation 1. $\square$

**Remark:** A main idea used in the proof is that only nontrivial irreps average to zero. This will be illustrated with examples in Section 4 below.

**Remark:** For finite groups, $N_0$ can be easily computed by averaging the trace of $\pi$, also known as the *character*, over $G$: $N_0 = \langle\text{Tr}(\pi(g))\rangle_{g \in G} = \text{Tr}(\langle\pi\rangle)$ (see Serre (2014)).

**Remark:** If the perceptron readout has a bias term $b$ i.e. the output of the perceptron is $w^\top \pi(g)r + b$, then this can be thought of as adding an invariant dimension. This is because the output can be written $\tilde{w}^\top \tilde{\pi}(g)\tilde{r}$ where $\tilde{w} = (w, b)$, $\tilde{r} = (r, 1)$, and $\tilde{\pi}(g) = \pi(g) \oplus 1$ is $\pi$ with an extra row and column added with a 1 in the last position and zeros everywhere else. Hence the fraction of separable dichotomies is $f(P, N_0 + 1)$.

These results extend immediately to a notion of VC-dimension of group-invariant perceptrons.

**Corollary 1.** *Let the VC dimension for $G$-invariant perceptrons with representation $\pi$, $N_{VC}^\pi$, denote the maximum number $P$, such that there exist $P$ anchor points $\{r^\mu\}_{\mu=1}^P$ so that $\{(\pi(g)r^\mu, y^\mu)\}_{\mu \in [P], g \in G}$ is separable for all possible binary dichotomies $\{y^\mu\}_{\mu \in [P]}$. Then $N_{VC}^\pi = dim(V_0)$.*

*Proof.* By theorem 1 and Equation 1, all possible dichotomies are realizable for $P \leq \dim(V_0)$ provided the points $\langle\pi\rangle r^\mu$ are in general position. $\square$

Our theory also extends immediately to subgroups. For example, strided convolutions are equivariant to subgroups of the regular representation.

**Corollary 2.** *A solution to the the $G$-invariant classification problem necessarily solves the $G'$-invariant problem where $G'$ is a subgroup of $G$.*

*Proof.* Assume that $y^\mu w^\top r(gx^\mu) > 0$ for $g \in G$. $G' \subseteq G \implies y^\mu w^\top r(gx^\mu) > 0 \, \forall g \in G'$. $\square$

Consequently, the $G'$-invariant capacity is always higher than the capacity for the $G$-invariant classification problem.

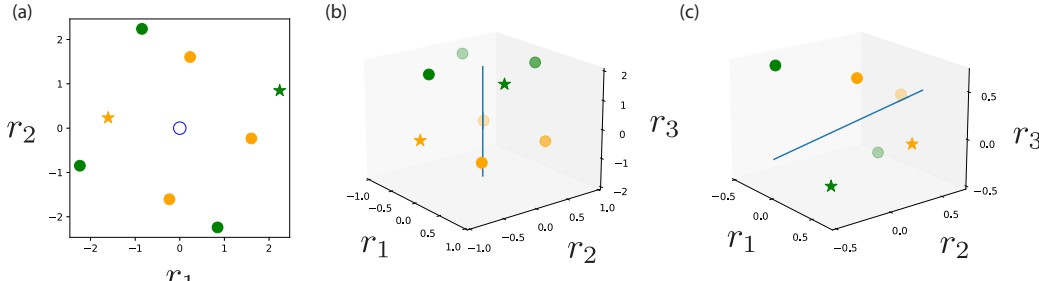

Figure 1: $\pi$-manifolds for different $\pi$, illustrating that only the fixed point subspace contributes to capacity. In each panel two manifolds are plotted, with color denoting class label. Stars indicate the random points $\boldsymbol{r}^\mu$ for $\mu \in \{1, 2\}$ where the orbits begin, and closed circles denote the other points in the $\pi$-manifold. For (a) and (b) the group being represented is $G = Z_4$ and for (c) $G = Z_3$. (a) Here $\pi(g)$ is the $2 \times 2$ rotation matrix $\boldsymbol{R}(2\pi g/4)$. The open blue circle denotes the fixed point subspace $\{\boldsymbol{0}\}$. (b) Here $\pi(g)$ is the $3 \times 3$ block-diagonal matrix with the first $2 \times 2$ block being $\boldsymbol{R}(2\pi g/4)$ and second $1 \times 1$ block being 1. The blue line denotes the fixed point subspace span$\{(0, 0, 1)\}$. (c) Here $\pi(g)$ is the $3 \times 3$ matrix that cyclically shifts entries of length-3 vectors by $g$ places. The blue line denotes the fixed point subspace span$\{(1, 1, 1)\}$.

## 4 EXAMPLE APPLICATION: THE CYCLIC GROUP $Z_m$

In this section we illustrate the theory in the case of the cyclic group $G = Z_m$ on $m$ elements. This group is isomorphic to the group of integers $\{0, 1, ..., m-1\}$ under addition modulo $m$, and this is the form of the group that we will consider. An example of this group acting on an object is an image that is shifted pixel-wise to the left and right, with periodic boundaries. In Appendix A.3 we show an application of our theory to the non-abelian Lie group $SO(3)$.

### 4.1 ROTATION MATRICES

The $2 \times 2$ discrete rotation matrices $\boldsymbol{R}(\theta_g) \equiv \begin{bmatrix} \cos(\theta_g) & -\sin(\theta_g) \\ \sin(\theta_g) & \cos(\theta_g) \end{bmatrix}$ where $\theta_g = 2\pi g/m$ and $g \in Z_m$, are one possible representation of $Z_m$; in this case $V = \mathbb{R}^2$. This representation is irreducible and nontrivial, which implies that the dimension of the fixed point subspace is 0 (only the origin is mapped to itself by $\boldsymbol{R}$ for all $g$). Hence the fraction of linearly separable dichotomies of the $\pi$-manifolds by Theorem 1 is $f(P, 0)$. This result can be intuitively seen by plotting the orbits, as in Figure 1a for $m = 4$. Here it is apparent that it is impossible to linearly separate two or more manifolds with different class labels, and that the nontrivial irrep $\boldsymbol{R}$ averages to the zero matrix.

The representation can be augmented by appending trivial irreps, defining $\pi : G \rightarrow \mathrm{GL}(\mathbb{R}^N)$ by $\pi(g) = \boldsymbol{R}(\theta_g) \oplus \boldsymbol{I} \equiv \begin{bmatrix} \boldsymbol{R}(\theta_g) & 0 \\ 0 & \boldsymbol{I} \end{bmatrix}$ where $\boldsymbol{I}$ is an $(N-2) \times (N-2)$-dimensional identity matrix. The number of trivial irreps is $N - 2$, so that the capacity is $f(P, N-2)$. This is illustrated in Figure 1b for the case $N = 3$. Here we can also see that the trivial irrep, which acts on the subspace span$\{(0, 0, 1)\}$, is the only irrep in the decomposition of $\pi$ that does not average to zero. This figure also makes intuitive the result of Lemma 1 that dichotomies are realizable on the $\pi$-manifolds if and only if the dichotomies are realizable on the centroids of the manifolds.

### 4.2 THE REGULAR REPRESENTATION OF $Z_m$

Suppose $\pi : Z_m \rightarrow \mathrm{GL}(V)$ is the representation of $Z_m$ consisting of the cyclic shift permutation matrices (this is called the **regular representation** of $Z_m$). In this case $V = \mathbb{R}^m$ and $\pi(g)$ is the matrix that cyclically shifts the entries of a length-$m$ vector $g$ places. For instance, if $m = 3$ and $\boldsymbol{v} = (1, 2, 3)$ then $\pi(2)\boldsymbol{v} = (2, 3, 1)$.

In Appendix A.2 we derive the irreps of this representation, which consist of rotation matrices of different frequencies. There is one copy of the trivial irrep $\pi_0(g) \equiv 1$ corresponding with the fixed point subspace span$\{\boldsymbol{1}_m\}$ where $\boldsymbol{1}_m$ is the length-$m$ all-ones vector. Hence the fraction of separable

dichotomies is $f(P, 1)$. This is illustrated in Figure 1c in the case where $m = 3$. The average of the regular representation matrix is $\langle \pi \rangle = \frac{1}{|G|} \mathbf{1}_m \mathbf{1}_m^\top$, indicating that $\langle \pi \rangle$ projects data along $\mathbf{1}_m$.

### 4.3 DIRECT SUMS OF REGULAR REPRESENTATIONS

For our last example we define a representation using the isomorphism $Z_m \cong Z_{m_1} \oplus Z_{m_2}$ for $m = m_1 m_2$ and $m_1$ and $m_2$ coprime[6]. Let $\pi^{(1)} : Z_{m_1} \to \mathrm{GL}(\mathbb{R}^{m_1})$ and $\pi^{(2)} : Z_{m_2} \to \mathrm{GL}(\mathbb{R}^{m_2})$ be the cyclic shift representations (i.e. the regular representations) of $Z_{m_1}$ and $Z_{m_2}$, respectively. Consider the representation $\pi^{(1)} \oplus \pi^{(2)} : Z_m \to \mathrm{GL}(\mathbb{R}^{m_1+m_2})$ defined by $(\pi^{(1)} \oplus \pi^{(2)})(g) \equiv \pi^{(1)}(g \mod m_1) \oplus \pi^{(2)}(g \mod m_2)$, the block-diagonal matrix with $\pi^{(1)}(g \mod m_1)$ being the first and $\pi^{(2)}(g \mod m_2)$ the second block.

There are two copies of the trivial representation in the decomposition of $\pi^{(1)} \oplus \pi^{(2)}$, corresponding to the one-dimensional subspaces $\mathrm{span}\{(\mathbf{1}_{m_1}, \mathbf{0}_{m_2})\}$ and $\mathrm{span}\{(\mathbf{0}_{m_1}, \mathbf{1}_{m_2})\}$, where $\mathbf{0}_{m_1}$ is the length-$k$ vector of all zeros. Hence the fraction of separable dichotomies is $f(P, 2)$. This reasoning extends simply to direct sums of arbitrary length $\ell$, yielding a fraction of $f(P, \ell)$.[7] These representations are used to build and test a novel G-convolutional layer architecture with higher capacity than standard CNN layers in Section 5.

These representations are analogous to certain formulations of grid cell representations as found in entorhinal cortex of rats (Hafting et al., 2005), which have desirable qualities in comparison to place cell representations (Sreenivasan & Fiete, 2011).[8] Precisely, a collection $\{Z_{m_k} \times Z_{m_k}\}_k$ of grid cell modules encodes a large 2-dimensional spatial domain $Z_m \times Z_m$ where $m = \prod_k m_k$.

## 5 G-EQUIVARIANT NEURAL NETWORKS

The proposed theory can shed light on the feature spaces induced by $G$-CNNs. Consider a single convolutional layer feature map for a finite[9] group $G$ with the following activation

$$a_{i,k}(\boldsymbol{x}) = \phi(\boldsymbol{w}_i^\top g_k^{-1} \boldsymbol{x}) , \ g_k \in G , \ i \in \{1, ..., N\} \tag{2}$$

for some nonlinear function $\phi$. For each filter $i$, and under certain choices of $\pi$, the feature map $\boldsymbol{a}_i(\boldsymbol{x}) \in \mathbb{R}^{|G|}$ exhibits the equivariance property $\boldsymbol{a}_i(g_k \boldsymbol{x}) = \pi(g_k) \boldsymbol{a}_i(\boldsymbol{x})$. We will let $\boldsymbol{a}(x) \in \mathbb{R}^{|G|N}$ denote a flattened vector for this feature map.

In a traditional periodic convolutional layer applied to inputs of width $W$ and length $L$, the feature map of a single filter $\boldsymbol{a}_i(\boldsymbol{x}) \in \mathbb{R}^{|G|}$ is equivariant with the regular representation of the group $G = Z_W \times Z_L$ (the representation that cyclically shifts the entries of $W \times L$ matrices). Here the order of the group is $|G| = WL$. Crucially, our theory shows that this representation contributes exactly one trivial irrep per filter (see Appendix A.4.1). Since the dimension of the entire collection of $N$ feature maps $\boldsymbol{a}(x)$ is $N|G|$ for finite groups $G$, one might naively expect capacity to be $P \sim 2N|G|$ for large $N$ from Gardner (1987). However, Theorem 1 shows that for $G$-invariant classification, only the trivial irreps contribute to the classifier capacity. Since the number of trivial irreps present in the representation is equal to the number of filters $N$, we have $P \sim 2N$.

We show in Figure 2 that our prediction for $f(P, N)$ matches that empirically measured by training logistic regression linear classifiers on the representation. We perform this experiment on both (a) a random convolutional network and (b) VGG-11 (Simonyan & Zisserman, 2015) pretrained on CIFAR-10 (Krizhevsky, 2009) by (liukuang, 2017). For these models we vary $\alpha$ by fixing the number of input samples and varying the number of output channels by simply removing channels from the output tensor. The convolutions in these networks are modified to have periodic boundary conditions while keeping the actual filters the same – see Appendix A.4.1 and Figure A.1 for more information and the result of using non-periodic convolutions, which impact the capacity but not the overall scaling with $N_0$.

---

[6]Two numbers are coprime if they have no common prime factor.

[7]Our results do not actually require that the $m_k$ be coprime, but rather that none of the $m_k$ divide one of the others. To see this, take $\tilde{m}$ and $\tilde{m}_k$, to be $m$ and the $m_k$ after being divided by all divisors common among them. Then $Z_{\tilde{m}} \cong \oplus_{k=1}^\ell Z_{\tilde{m}_k}$ and provided none of the $\tilde{m}_k$ are 1, one still gets a fraction of $f(P, \ell)$.

[8]Place cells are analogous to standard convolutional layers.

[9]We consider finite groups here for simplicity, but the theory extends to compact Lie groups.

Other GCNN architectures can have different capacities. For instance, a convolutional layer equivariant to the direct sum representation of Section 4.3 has double the capacity with $P \sim 4N$ (Figure 2c), since each output channel contributes two trivial irreps. See Appendix A.4.4 for an explicit construction of such a convolutional layer and a derivation of the capacity.

## 5.1 POOLING OPERATIONS

In convolutional networks, local pooling is typically applied to the feature maps which result from each convolution layer. In this section, we describe how our theory can be adapted for codes which contain such pooling operations. We will first assume that $\pi$ is an $N$-dimensional representation of $G$. Let $\mathcal{P}(\boldsymbol{r}) : \mathbb{R}^N \to \mathbb{R}^{N/k}$ be a pooling operation which reduces the dimension of the feature map. The condition that a given dichotomy $\{y^\mu\}$ is linearly separable on a pooled code is

$$\exists \boldsymbol{w} \in \mathbb{R}^{N/k} \, \forall \mu \in [P], g \in G : y^\mu \boldsymbol{w}^\top \mathcal{P}(\pi(g)\boldsymbol{r}^\mu) > 0 \tag{3}$$

We will first analyze the capacity when $\mathcal{P}(\cdot)$ is a linear function (average pooling) before studying the more general case of local non-linear pooling on one and two-dimensional signals.

### 5.1.1 LOCAL AVERAGE POOLING

In the case of average pooling, the pooling function $\mathcal{P}(\cdot)$ is a linear map, represented with matrix $\boldsymbol{P}$ which averages a collection of feature maps over local windows. Using an argument similar to Lemma 1 (Lemma 2 and Theorem 2 in Appendix A.4.2), we prove that the capacity of a standard CNN is not changed by local average pooling: for a network with $N$ filters, local average pooling preserves the one trivial dimension for each of the $N$ filters. Consequently the fraction of separable dichotomies is $f(P, N)$.

### 5.1.2 LOCAL NONLINEAR POOLING

Often, nonlinear pooling operations are applied to downsample feature maps. For concreteness, we will focus on one-dimensional signals in this section and relegate the proofs for two-dimensional signals (images) to Appendix A.4.2. Let $\boldsymbol{r}(\boldsymbol{x}) \in \mathbb{R}^{N \times D}$ represent a feature map with $N$ filters and length-$D$ signals. Consider a pooling operation $\mathcal{P}(\cdot)$ which maps the $D$ pixels in each feature map into new vectors of size $D/k$ for some integer $k$. Note that the pooled code is equivariant to the subgroup $H = \mathbb{Z}^{D/k}$, in the sense that $\mathcal{P}(\pi(h)\boldsymbol{r}) = \rho(h)\mathcal{P}(\boldsymbol{r})$ for any $h \in H$. The representation $\rho$ is the regular representation of the subgroup $H$. We thus decompose $G$ into cosets of size $D/k$: $g = jh$, where $j \in Z_k$ and $h \in Z_{D/k}$. The condition that a vector $\boldsymbol{w}$ separates the dataset is

$$\forall \mu \in [P], j \in Z_k, h \in H : y^\mu \boldsymbol{w}^\top \rho(h)\mathcal{P}(\pi(j)\boldsymbol{r}^\mu) > 0. \tag{4}$$

We see that there are effectively $k$ points belonging to each of the $P$ orbits in the pooled code. Since $\mathcal{P}(\cdot)$ is nonlinear, the averaging trick utilized in Lemma 1 is no longer available. However, we can obtain a lower bound on the capacity, $f(P, \lfloor N/k \rfloor)$, from a simple extension of Cover's original proof technique as we show in Appendix A.4.3. Alternatively, an upper bound $f \leq f(P, N)$ persists since a $\boldsymbol{w}$ which satisfies Equation 4 must separate $\langle \rho(h) \rangle_{h \in H} \mathcal{P}(\boldsymbol{r}^\mu)$. This upper bound is tight when all $k$ points $\langle \rho(h) \rangle_{h \in H} \mathcal{P}(\pi(j)\boldsymbol{r}^\mu)$ coincide for each $\mu$, giving capacity $f(P, N)$. This is what occurs in the average pooling case where the upper bound $f \leq f(P, N)$ is tight. Further, if we are only interested in the $H$-invariant capacity problem, the fraction of separable dichotomies is $f(P, N)$, since $\rho$ is a regular representation of $H$ as we show in 4.

### 5.1.3 GLOBAL POOLING

Lemma 1 shows that linear separability of $\pi$-manifolds is equivalent to linear separability after global average pooling (i.e. averaging the representation over the group). This is relevant to CNNs, which typically perform global average pooling following the final convolutional layer, which is typically justified by a desire to achieve translation invariance. This strategy is efficient: Lemma 1 implies that it allows the learning of optimal linear readout weights following the pooling operation, given only a single point from each $\pi$-manifold (i.e. no need for data augmentation). Global average pooling reduces the problem of linearly classifying $|G|P$ points in a space of dimension $N$ to that of linearly classifying $P$ points in a space of dimension $\dim V_0$. Note that by an argument similar to Section 5.1.2, global max pooling may in contrast reduce capacity.

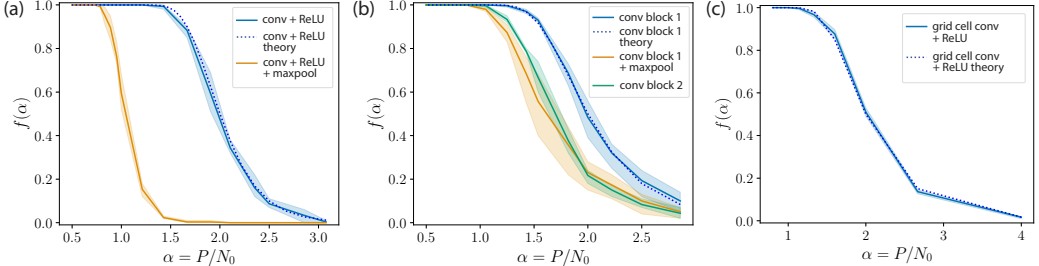

Figure 2: Capacity of GCNN representations. Solid lines denote the empirically measured fraction $f(\alpha)$ of 100 random dichotomies for which a logistic regression classifier finds a separating hyperplane, where $\alpha = P/N_0$. Dotted lines denote theoretical predictions. Shaded regions depict 95% confidence intervals over random choice of inputs, as well as network weights in (a) and (c). (a) $f(\alpha)$ of a random periodic convolutional layer after ReLU (blue line) and followed by 2x2 max pool (orange line), with $P = 40$ and $N_0 = \#$ output channels. Max pooling reduces capacity by a factor between 1/4 and 1 as predicted by our theory. (b) $f(\alpha)$ of VGG-11 pretrained on CIFAR-10 after a periodic convolution, batchnorm, and ReLU (blue line), followed by a 2x2 maxpool (orange line), and then another set of convolution, batchnorm, and ReLU (green line), with $P = 20$ and $N_0 = \#$ output channels. Max pooling reduces capacity as predicted. (c) $f(\alpha)$ after a random convolutional layer equivariant to the direct sum representation of $Z_{10} \oplus Z_8$ as defined in Section 4.3, with $P = 16$ and $N_0 = 2$ (# output channels).

## 5.2 INDUCED REPRESENTATIONS

Induced representations are a fundamental ingredient in the construction of general equivariant neural network architectures (Cohen et al., 2019b). Here we state our result, and relegate a formal definition of induced representations and the proof of the result to Appendix A.4.5.

**Proposition 1.** *Let $\pi$ be a representation of a finite group induced from $\rho$. Then the fraction of separable dichotomies of $\pi$-manifolds is equal to that of the $\rho$-manifolds.*

## 6 DISCUSSION AND CONCLUSION

Equivariance has emerged as a powerful framework to build and understand representations that reflect the structure of the world in useful ways. In this work we take the natural step of quantifying the expressivity of these representations through the well-established formalism of perceptron capacity. We find that the number of "degrees of freedom" available for solving the classification task is the dimension of the space that is fixed by the group action. This has the immediate implication that capacity scales with the number of output channels in standard CNN layers, a fact we illustrate in simulations. However, our results are also very general, extending to virtually any equivariant representation of practical interest – in particular, they are immediately applicable to GCNNs built around more general equivariance relations, as we illustrate with an example of a GCNN equivariant to direct sum representations. We also calculate the capacity of induced representations, a standard tool in building GCNNs, and show how local and global pooling operations influence capacity. The measures of expressivity explored here could prove valuable for ensuring that models with general equivariance relations have high enough capacity to support the tasks being learned, either by using more filters or by using representations with intrinsically higher capacity. We leave this to future work.

While the concept of perceptron capacity has played an essential role in the development of machine learning systems (Schölkopf & Smola, 2002; Cohen et al., 2020) and the understanding of biological circuit computations (Brunel et al., 2004; Chapeton et al., 2012; Rigotti et al., 2013; Brunel, 2016; Rubin et al., 2017; Pehlevan & Sengupta, 2017; Lanore et al., 2021; Froudarakis et al., 2020), more work is wanting in linking it to other computational attributes of interest such as generalization. A comprehensive picture of the computational attributes of equivariant or otherwise structured representations of artificial and biological learning systems will likely combine multiple measures, including perceptron capacity. There is much opportunity, and indeed already significant recent work (e.g. Sokolic et al. (2017)).

## REPRODUCIBILITY STATEMENT

Code for the simulations can be found at `https://github.com/msf235/group-invariant-perceptron-capacity`. This code includes an environment.yml file that can be used to create a python environment identical to the one used by the authors. The code generates all of the plots in the paper.

While the main text is self-contained with the essential proofs, proofs of additional results and further discussion can be found in the Appendices below. These include

**Appendix A.1** a glossary of definitions and notation.

**Appendix A.2** a derivation of the irreps for the regular representation of the cyclic group $Z_m$.

**Appendix A.3** a description of the irreps for the group $SO(3)$ and the resulting capacity.

**Appendix A.4** a description of the construction of the GCNNs used in the paper, the methods for empirically measuring the fraction of linearly separable dichotomies, a description of local pooling, and complete formal proofs of the fraction of linearly separable dichotomies for these network representations (including with local pooling). This appendix also includes more description of the induced representation and a formal proof of the fraction of linearly separable dichotomies.

Appendix A.4.1 also contains an additional figure, Figure A.1.

## ACKNOWLEDGEMENTS

MF and CP are supported by the Harvard Data Science Initiative. MF thanks the Swartz Foundation for support. BB acknowledges the support of the NSF-Simons Center for Mathematical and Statistical Analysis of Biology at Harvard (award #1764269) and the Harvard Q-Bio Initiative. ST was partially supported by the NSF under grant No. DMS-1439786.

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

# A    APPENDIX

## A.1    NOTATION AND GLOSSARY

- $\boldsymbol{x}$: an abstract notation for an input object
- $\boldsymbol{r}(\boldsymbol{x})$: a feature map of the input to an $N$ dimensional vector space.
- $\pi$: $N$ dimensional linear representation of group $G$. For each $g \in G$, $\pi(g) \in GL(\mathbb{R}^N)$ is an $N \times N$ invertible real matrix.
- Equivariance property: $\boldsymbol{r}(g\boldsymbol{x}) = \pi(g)\boldsymbol{r}(\boldsymbol{x})$ for all $g \in G$ and all $\boldsymbol{x}$.
- Invariant measure: a measure $\mu : G \to \mathbb{R}_+$ on $G$ with $\mu(gS) = \mu(S) = \mu(Sg)$. For finite groups, the uniform distribution. For locally compact topological groups, the Haar measure.
- $\langle \cdot \rangle_{g \in G}$: an average over the invariant measure of $G$
- Irreducible representation (irrep): an irreducible representation $\rho$ on vector space $V$ satisfies $\rho(g)v \in V$ for all $v \in V, g \in G$.
- Character $\chi(g)$: the trace of the representation $\chi(g) = \text{Tr } \pi(g)$.
- Fixed point subspace: the subspace $V_0$ for which $\pi(g)v \in V_0$ for all $v \in V_0$.
- General position: a collection of $P$ points in general position in an $N$ dimensional vector space have the property that any subset of $k \leq N$ points are linearly independent. These points are generic in the sense that they satisfy no more linear relationships than they must.
- Dichotomy: a particular binary labeling $\{y^\mu\}$ of $P$ points $\{\boldsymbol{x}^\mu\}$.
- $f(P, N)$: fraction of linearly separable dichotomies given by Cover's function counting theorem (Cover, 1965).
- VC dimension: the largest possible integer $P$ such that there exist $P$ points where all possible dichotomies $\{y^\mu\}$ can be realized by the model Abu-Mostafa et al. (2012).
- Capacity: the largest possible ratio $\alpha_c = P/N$ where $P$ points in general position can be linearly separated by a $N$ dimensional perceptron with probability 1 in an asymptotic limit where $P, N \to \infty$ with $P/N = O_{N,P}(1)$. The classical result is $\alpha_c = 2$ (Gardner, 1987; Shcherbina & Tirozzi, 2003).
- $\mathcal{P}(\cdot)$: a local pooling operation.

## A.2    IRREPS FOR THE CYCLIC GROUP

Here we compute the irreps of representations $\pi : Z_m \to \text{GL}(V)$ of the cyclic group $Z_m$ over a real vector space $V$ (see (Serre, 2014) for a derivation of the irreps when $V$ is a complex vector space). To find the irreps, one can use the form for the eigenvalues and eigenvectors for circulant matrices, since all the $\pi(g)$ are circulant. This results in the simultaneous diagonalization $\pi(g) = \boldsymbol{V} \left(1 \oplus \boldsymbol{R}(2\pi g/m) \oplus \boldsymbol{R}(4\pi g/m) \oplus \cdots \oplus \boldsymbol{R}((m-1)\pi g/m)\right) \boldsymbol{V}^\top$ where $\boldsymbol{V}$ is the real-valued version of the discrete Fourier transform matrix (the columns are proportional to cosines and sines of varying frequencies, along with a column proportional to $\boldsymbol{1}_m$).

Note that the 2x2 rotation matrices $\boldsymbol{R}(2\pi gk/m)$ are irreps for $k \neq m/2$ and $k \neq 0$, since there is no one-dimensional subspace of $\mathbb{R}^2$ that is invariant to $\boldsymbol{R}(2\pi gk/m)$ for all $g$. The exception for $k = m/2$ if $m$ is even, gives $\boldsymbol{R}(2\pi gk/m) = (-1)^g \boldsymbol{I}$, which corresponds to rotation of 180 degrees. The subspace $\text{span}\{(1, 0)\}$ is invariant to this action, so that $\boldsymbol{R}(2\pi gk/m)$ is not an irrep. This representation can thus be reduced to an action on a one-dimensional subspace, represented by $(-1)^g$. The case $k = 0$ gives the trivial representation.

## A.3    SO(3): A NON-ABELIAN LIE GROUP

The special orthogonal group SO(3) on 3 dimensions (rotation group), the $3 \times 3$ orthogonal matrices with determinant $+1$, can also be analyzed within our theory. G-convolutional neural networks that are equivariant to SO(3) rotations have become of high interest in the physical sciences and computer vision where the objects of interest often respect these symmetries (Anderson et al.,

2019; Cohen et al., 2018; Esteves et al., 2018; Kondor et al., 2018a) The irreducible representations have the form $\boldsymbol{B}_{k_m}$ where $\boldsymbol{B}_{k_m}$ are $(2k_m + 1) \times (2k_m + 1)$ block matrices, known as Wigner $D$-matrices (Wigner, 1931). The trivial irreps correspond to the one-dimensional irreps with $k = 0$. Thus, the SO(3)-invariant classification capacity merely counts the number of trivial irreps which have $N_0 = \sum_m \delta_{k_m,0}$. The capacity is again $f(P, N_0)$.

## A.4 G-EQUIVARIANT CONVOLUTIONAL LAYERS

### A.4.1 STANDARD CONVOLUTIONAL LAYERS

A convolutional layer consists of a set of $N$ $k \times k'$ filters $F_i$ that are convolved (technically cross-correlated) with a stack of $M$ $W \times L$ input tensors. Here $M$ is the number of input channels and $N$ the number of output channels. The convolution runs each filter (i.e. takes the dot product at all possible positions) over each of the $W \times L$ input tensors, and the result is averaged across the $M$ input channels to produce the output of one output channel. In the positions where the filters approach the edges of the input tensor, different choices can be made about how to handle these edge conditions. The standard choice is to pad the edges with some number of zeros depending on the desired shape of the output tensor and run the convolution out to the end of the padded image. Another possible choice is to loop the edges of the input tensor together, so that the filter is applied to the other side of the input tensor as it runs off the edge. This periodic boundary condition allows us to write the convolution formally in terms of group actions, and to apply our theory directly. When convolutions are not periodic, the resulting capacity increases somewhat but still follows the $P/N_0$ scaling of the periodic convolutions (Figure A.1).

For the random convolutional layers of Figure 2a, the input tensors are size $10 \times 10$ and the number of input channels are $M = 3$, as for standard color images. Each entry of these tensors is normally distributed with mean 0. The filters are also of size $10 \times 10$ with periodic boundary conditions, and are initialized according to a normal Xavier distribution with parameters that are the default for Pytorch 1.9. The convolution is implemented via the Pytorch 1.9 implementation of Conv2d with padding_mode="circular" and padding=0 in the case of periodic boundary conditions. The bias term of the convolution is set to zero and the convolution is followed by a ReLU nonlinearity (blue line). The resulting figures do not change appreciably for different choices of input tensor size, number of input channels, or size of filters (though note that the nonlinearity is essential for satisfying the general position condition of Theorem 1; otherwise, the capacity would be determined by the number of input channels rather than number of output channels). The output of this convolution is then fed through a $2 \times 2$ max pooling layer (orange line in Figure 2a), provided by Pytorch 1.9's MaxPool2d.

The pretrained VGG-11 layers used in Figure 2b and Figure A.1 are taken from liukuang (2017). The first convolutional block (blue line) consists of $3 \times 3$ pretrained filters applied to CIFAR-10 image tensors randomly selected from the validation set and normalized in the same way they are normalized during training (see liukuang (2017) for details). These images are of size $32 \times 32$ and have $M = 3$ input channels, followed by a batch normalization layer in evaluation mode (fixed parameters), and then followed by a ReLU nonlinearity. The boundary conditions of these convolutions are set to be periodic in Figure 2b and nonperiodic with a zero padding sizes of 1 in Figure A.1, and the bias term is set to zero. The batch normalization is an element-wise operation and so equivariant to the representations we consider – thus this operation is not expected and is not observed to affect the perceptron capacity. This convolutional block is then followed by a $2 \times 2$ max pooling layer (orange line). Finally, another convolutional block of $3 \times 3$ filters, batch normalization, and ReLU nonlinearity are applied (green line).

The fraction of linearly separable dichotomies is measured empirically by using the scikit-learn LogisticRegression implementation of logistic regression, with a tolerance value of tol=1e-18 and an inverse regularization value of C=1e8. The maximum number of iterations is set to 500. An intercept (i.e. bias) is not used for this fit.

To formally prove that the fraction of separable dichotomies is $f(P, N)$ for standard periodic convolutional layers, first note that the convolution is equivariant with respect to cyclic permutation of the inputs and of the outputs. The representation for cyclically permutating the output tensor can be written $\bigoplus_{k=1}^{N} \pi$ where $\pi$ is the representation that cyclically permutes the entries of $W \times L$ matrices. Since each copy of $\pi$ contains one trivial irrep in its decomposition into a direct sum of

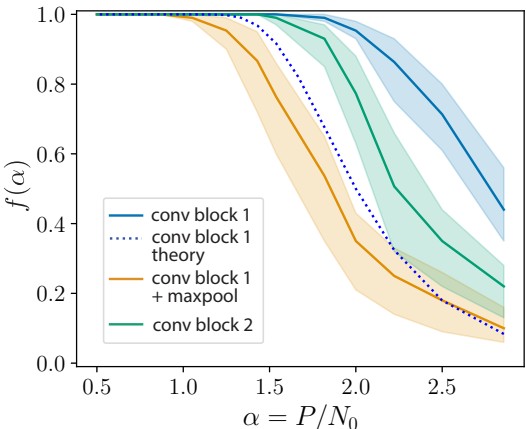

Figure A.1: The fraction of realizable dichotomies of non-periodic convolutional layers is higher than periodic convolutional layers, but still obeys the same scaling. Details are exactly as in Figure 2b, but using non-periodic convolutions with a zero padding of size 1. Here the theory line refers to the theory for periodic convolutions.

irreps, the direct sum $\bigoplus_{k=1}^{N} \pi$ contains $N$ trivial irreps in its decomposition. The final step to use Theorem 1 is to argue that the centroids of the manifolds are in general position. Since a nonlinearity (ReLU) is applied to the output of the convolution, and since there is no particular structure in the convolutional filters beyond possibly sparsity, we can generically expect this to be the case.

### A.4.2 Local Pooling

First we prove an extension of Lemma 1 to equivariant linear maps. This will be used to show that average pooling does not affect the capacity of the regular representation of $Z_m$.

**Lemma 2.** *Let $\pi$ be a representation of the group $G$ and suppose the matrix $\boldsymbol{P}$ is equivariant with respect to the restriction of $\pi$ to a subgroup $H \subseteq G$, so that for all $h \in H$ $\boldsymbol{P}\pi(h) = \rho(h)\boldsymbol{P}$ for some representation $\rho$ of $H$. Let $R$ denote a set of representatives of $G/H$. Then we have the following equivalence.*

$$\exists \boldsymbol{w} \; \forall \mu \in [P], g \in G : y^{\mu} \boldsymbol{w}^{\top} \boldsymbol{P}\pi(g)\boldsymbol{r}^{\mu} > 0$$
$$\iff \exists \boldsymbol{w} \; \forall \mu \in [P] \; \forall g' \in R \; : \; y^{\mu} \boldsymbol{w}^{\top} \boldsymbol{P}\langle\pi(h)\rangle_{h \in H}\pi(g')\boldsymbol{r}^{\mu} > 0.$$

*Proof.* For the forward implication, we write the coset decomposition $g = hg'$ of $g$ and average over $H$ to find

$$\forall g \in G : y^{\mu} \boldsymbol{w}^{\top} \boldsymbol{P}\pi(g)\boldsymbol{r}^{\mu} > 0 \iff \forall h \in H, g' \in R : y^{\mu} \boldsymbol{w}^{\top} \boldsymbol{P}\pi(h)\pi(g')\boldsymbol{r}^{\mu} > 0$$
$$\implies \forall g' \in R : y^{\mu} \boldsymbol{w}^{\top} \boldsymbol{P}\langle\pi(h)\rangle_{h \in H}\pi(g')\boldsymbol{r}^{\mu} > 0.$$

For the backward implication, suppose $y^{\mu} \boldsymbol{w}^{\top} \boldsymbol{P}\langle\pi(h)\rangle_{h \in H}\pi(g')\boldsymbol{r}^{\mu} > 0$ for all representatives $g' \in R$, and define $\tilde{\boldsymbol{w}} = \langle\rho(h)\rangle_{h \in H}^{\top}\boldsymbol{w}$. For any $g \in G$, take a coset decomposition $g = hg'$ for $h \in H$ and $g' \in R$. We then have

$$y^{\mu}\tilde{\boldsymbol{w}}^{\top}\boldsymbol{P}\pi(g)\boldsymbol{r}^{\mu} = y^{\mu}\tilde{\boldsymbol{w}}^{\top}\boldsymbol{P}\pi(h)\pi(g')\boldsymbol{r}^{\mu} \quad \text{(Coset decomposition)}$$
$$= y^{\mu}\tilde{\boldsymbol{w}}^{\top}\rho(h)\boldsymbol{P}\pi(g')\boldsymbol{r}^{\mu} \; (\boldsymbol{P} \text{ is } H\text{-equivariant})$$
$$= y^{\mu}\boldsymbol{w}^{\top}\langle\rho(h')\rangle_{h' \in H}\rho(h)\boldsymbol{P}\pi(g')\boldsymbol{r}^{\mu} \text{ (Definition of } \tilde{\boldsymbol{w}})$$
$$= y^{\mu}\boldsymbol{w}^{\top}\langle\rho(h)\rangle_{h \in H}\boldsymbol{P}\pi(g')\boldsymbol{r}^{\mu} \text{ (Invariance of measure)}$$
$$= y^{\mu}\boldsymbol{w}^{\top}\boldsymbol{P}\langle\pi(h)\rangle_{h \in H}\pi(g')\boldsymbol{r}^{\mu} \; (\boldsymbol{P} \text{ is linear and } H\text{-equivariant})$$
$$> 0 \text{ (By assumption).} \tag{5}$$

The implication follows. $\qquad\square$

**Lemma 3.** *For the regular representation of $G = Z_D$, a local average pooling over windows of size $k$ generates a matrix $\boldsymbol{P}$ which is equivariant with respect to the subgroup $H = Z_{D/k}$ with the property that*

$$\boldsymbol{P}\langle\pi(h)\rangle_{h\in H} = a\boldsymbol{P}\langle\pi(g)\rangle_{g\in G} \tag{6}$$

*where $a > 0$ is a positive constant.*

*Proof.* First, note that the new pooled code is the regular representation of $H$ since shifts of size $mk$ in the original feature map corresponds to shifts of length $m$ in the pooled code. Thus $\boldsymbol{P}$ is equivariant to $H = Z_{D/k}$. Next we note the following two facts

$$\boldsymbol{P}\langle\pi(h)\rangle_{h\in H} = a'\boldsymbol{1}_{D/k}\boldsymbol{1}_D^\top \tag{7}$$

$$\boldsymbol{P}\langle\pi(h)\rangle_{g\in G} = a''\boldsymbol{1}_{D/k}\boldsymbol{1}_D^\top \tag{8}$$

$a'$ and $a''$ are positive constants and $\boldsymbol{1}_D$ and $\boldsymbol{1}_{D/k}$ are the $D$ and $D/k$ dimensional vector of all ones, respectively. Thus, we have that $\boldsymbol{P}\langle\pi(h)\rangle_{h\in H} = a\boldsymbol{P}\langle\pi(g)\rangle_{g\in G}$ where $a$ is a positive constant. □

**Theorem 2.** *The fraction of linearly separable dichotomies of a CNN pooling layer with $N$ filters after average pooling from feature maps with the size of the input image $W \times L$ to pooled feature maps of size $W/k \times L/k$ is $f(P, N)$, i.e. no capacity is lost due to local average pooling.*

*Proof.* The CNN layer before pooling is a regular representation $\pi$ of the full group $G = Z_W \times Z_L$, applied to each of the $M$ input channels via a direct a sum $\bigoplus_{j=1}^M \pi$. The layer after pooling is a regular representation $\rho$ of the subgroup $H = Z_{W/k} \times Z_{L/k}$, also applied to the output channels via a direct sum $\bigoplus_{j=1}^N \rho$. Let $R$ be a set of representatives of $G/H$. Since $\boldsymbol{P}$ is equivariant to $\pi$ and $\rho$ over $H$ we have by the previous two lemmas that

$$\forall g \in G : y^\mu \boldsymbol{w}^\top \boldsymbol{P}\pi(g)\boldsymbol{r}^\mu > 0 \iff \forall g' \in R : y^\mu \boldsymbol{w}^\top \boldsymbol{P}\langle\pi(h)\rangle_{h\in H}\pi(g')\boldsymbol{r}^\mu > 0$$

$$\iff \forall g' \in R : y^\mu \boldsymbol{w}^\top \boldsymbol{P}\langle\pi(g)\rangle_{g\in G}\pi(g')\boldsymbol{r}^\mu > 0$$

$$\iff y^\mu \boldsymbol{w}^\top \boldsymbol{P}\langle\pi(g)\rangle_{g\in G}\boldsymbol{r}^\mu > 0$$

$$\iff y^\mu \boldsymbol{w}^\top \boldsymbol{P}\langle\pi(h)\rangle_{h\in H}\boldsymbol{r}^\mu > 0$$

$$\iff y^\mu \boldsymbol{w}^\top \langle\rho(h)\rangle_{h\in H}\boldsymbol{P}\boldsymbol{r}^\mu > 0$$

Thus the capacity is determined by the rank of $\langle\rho(h)\rangle_{h\in H}$, assuming the $\langle\rho(h)\rangle_{h\in H}\boldsymbol{P}\boldsymbol{r}^\mu$ are in general position. Since each $\rho$ is a copy of the regular representation for $H$, the rank of $\langle\bigoplus_{j=1}^N \rho(h)\rangle_{h\in H}$ is merely $N$. Thus the fraction of linearly separable dichotomies is $f(P, N)$, the same as the capacity before pooling. □

Now we prove local pooling operations in a standard CNN preserve a regular representation of a subgroup of the cyclic group.

**Lemma 4.** *Suppose $P$ is a local pooling operation on two-dimensional signals (CNN feature maps), and that $\pi$ is the regular representation of a group $G = Z_W \times Z_L$ on code $\boldsymbol{a}(\boldsymbol{x})$. A pooled feature map $\boldsymbol{r} = \mathcal{P}(\boldsymbol{a})$ which acts on $k \times k$ windows of $\boldsymbol{a}$ is a regular representation of the subgroup $H = Z_{W/k} \times Z_{L/k}$.*

*Proof.* Suppose an equivariant feature map $\boldsymbol{a}(\boldsymbol{x}) \in \mathbb{R}^{W \times L \times N}$ has corresponding regular representation of the group $G = Z_W \times Z_L$ for each of the $N$ filters. Consider any local pooling operation $\mathcal{P}(\cdot)$ (such as average or maximum) which acts on $k \times k$ patches where $k$ divides both $W$ and $L$.

$$r_{ij,h}(x) = \mathcal{P}\left(\{a_{i',j',h}(x) \mid i' \in [ik, (i+1)k], j' \in [jk, (j+1)k]\}\right) \tag{9}$$

Note that for $k > 1$, $\boldsymbol{r}(x)$ is no longer equivariant to $G$ since the representation does not satisfy the homomorphism property for shifts with length $\ell$ not divisible by $k$. However, the new code is equivariant to a subgroup $H = Z_{W/k} \times Z_{L/k}$, namely vertical and horizontal shifts with length divisible by $k$. Let $\boldsymbol{\pi}_{nk,mk}^{\boldsymbol{x}}$ represent a vertical shift of the image $\boldsymbol{x}$ by $nk$ pixels and horizontal shift

by $mk$ pixels. Note that $a_{ij,h}(\pi^{\boldsymbol{x}}_{nk,mk}\boldsymbol{x}) = a_{i+nk,j+mk}(\boldsymbol{x})$ since $\boldsymbol{a}(\boldsymbol{x})$ is equivariant. Then, the $h$-th pooled feature map transforms as

$$
\begin{aligned}
r_{ij,h}(\pi^{\boldsymbol{x}}_{nk,mk}\boldsymbol{x}) &= \mathcal{P}\left(\{a_{i',j',h}(\pi^{\boldsymbol{x}}_{nk,mk}x) \mid i' \in [ik,(i+1)k], j' \in [jk,(j+1)k]\}\right) \\
&= \mathcal{P}\left(\{a_{i'+nk,j'+mk,h}(\boldsymbol{x}) \mid i' \in [ik,(i+1)k], j' \in [jk,(j+1)k]\}\right) \\
&\quad (\boldsymbol{a} \text{ is Equivariant to } \pi^x) \\
&= \mathcal{P}\left(\{a_{i',j',h}(\boldsymbol{x}) \mid i' \in [(n+i)k,(n+i+1)k], j' \in [(m+j)k,(m+j+1)k]\}\right), \\
&\quad (k \text{ divides } W, H) \\
&= r_{i+n,j+m,h}(\boldsymbol{x}) \, , \quad (\text{Definition of } \boldsymbol{r})
\end{aligned}
$$

We thus find that the code is a regular representation of the subgroup of the cyclic translations $H = \{(nk, mk)\}_{n \in [H/k], m \in [W/k]}$. This new group $G'$ has dimension $|H| = \frac{1}{k^2}|G|$. $\qquad\square$

### A.4.3 LOWER BOUND AND UPPER BOUND ON CAPACITY FOR NONLINEAR POOLING

**Theorem 3.** *Suppose a code which is equivariant to a finite group $G$ is pooled to a new code which is equivariant to a finite subgroup $H \subseteq G$. Suppose the number of trivial dimensions in the original $G$-equivariant code is $N_0$. Then the fraction of linearly separable dichotomies on the $G$-invariant problem for the pooled code is at least $f(P, \lfloor N_0/k \rfloor)$ where $k = |G/H|$. Similarly the fraction is at most $f(P, N_0)$.*

*Proof.* The pooled code, by assumption, has the property $\mathcal{P}(\pi(hg')\boldsymbol{r}) = \rho(h)\mathcal{P}\left(\pi(g')\boldsymbol{r}\right)$ for any $h \in H$ and $g' \in R$, where $R$ is a set of representatives of $G/H$. The $G$-invariant separability condition amounts to the proposition

$$\exists \boldsymbol{w} \forall \mu \in [P] \, \forall h \in H, g' \in R : y^\mu \boldsymbol{w}^\top \rho(h)\mathcal{P}(\pi(g')\boldsymbol{r}^\mu) > 0 \tag{10}$$

$$\iff \exists \boldsymbol{w} \forall \mu \in [P] \, \forall g' \in R : y^\mu \boldsymbol{w}^\top \langle \rho(h) \rangle_{h \in H} \mathcal{P}(\pi(g')\boldsymbol{r}^\mu) > 0; \tag{11}$$

in other words, a solution on the right hand side affords a solution over all of the manifolds generated in the input space. We see that, this requires considering if this particular dichotomy is linearly separble on the $Pk$ anchor points $\langle \rho(h) \rangle_{h \in H} \mathcal{P}(\pi(g')\boldsymbol{r}^\mu)$. The simplest strategy to obtain an upper bound is to consider what happens when a single new manifold is added. We see that when a single new base point $\boldsymbol{r}$ is added it corresponds to $k$ new points in the orbit $\langle \rho \rangle \mathcal{P}(\pi(g')\boldsymbol{r})$ for all $g' \in R$. Suppose that $\langle \rho(h) \rangle_{h \in H}$ has rank $N_0$. Let $C(P, N_0)$ represent the number of linearly separable dichotomies for $P$ $G$-orbits in $N_0$ trivial dimensions. Upon the addition of the $k$ new points ($P \to P+1$), we find that some of the pre-existing separable dichotomies give a new separable dichotomy. This can be guaranteed to occur when a $\boldsymbol{w}$ separates the old dichotomy and has $\boldsymbol{w}^\top \langle \rho \rangle \cdot \mathcal{P}(\pi(g')\boldsymbol{r}) = 0$ for the new anchor point $\boldsymbol{r}$ (but this condition is not *necessary* for a new dichotomy to be separable). This condition means that the original dichotomy is separable in the $N_0 - k$ dimensional subspace $\{\boldsymbol{w} : \boldsymbol{w} \cdot \mathcal{P}(\pi(g')\boldsymbol{r}) = 0\}$. By making infinitesimal adjustment to this $\boldsymbol{w}$ the correct label on this new orbit can be achieved without altering the labels on any other dichotomy. Since this argument gives a sufficient but not necessary condition to generate a new dichotomy, we obtain the following inequality

$$C(P+1, N_0) \geq C(P, N_0) + C(P, N_0 - k). \tag{12}$$

Solving this recursion gives the capacity $f_{Pooled}(P, N_0) \geq f_{Cover}(P, \lfloor N_0/k \rfloor)$. The greatest capacity occurs in the special case where $\langle \rho \rangle \mathcal{P}(\pi(g')\boldsymbol{r}) = \langle \rho \rangle \mathcal{P}(\boldsymbol{r})$. In this case, the usual counting theorem applies giving a fraction of separable dichotomies of $f(P, N_0)$. This is achieved, for instance in average pooling as we showed in Theorem 2. $\qquad\square$

### A.4.4 DIRECT SUM EQUIVARIANT CONVOLUTIONAL LAYERS

Here we describe how to build a convolutional layer architecture that is equivariant with respect to the regular representation in the input space and the direct sum representations introduced in Section 4.3 in the output space. For the following we assume that $m_1$ and $m_2$ are coprime, though see the footnote in Section 4.3 for a discussion of how to loosen this requirement.

The input data is a $W \times L \times M$ tensor where $M$ is the number of input channels. The first step is to simply take the output of a standard convolution (in our simulations we also apply a ReLU

nonlinearity) applied to this input with periodic boundary conditions, resulting in a $W \times L \times N$ tensor where $N$ is the number of output channels. The next step is to, for each of the output channels, take an average (or maximum) between entries spaced $m_1$ entries apart horizontally or vertically in the matrix, resulting in an $m_1 \times m_1$ matrix. In our simulations we took averages rather than maximums. This is repeated for the other number $m_2$, resulting in an $m_2 \times m_2$ matrix. This is repeated for every output channel, resulting in $N$ matrices of size $m_1 \times m_1$ and $N$ matrices of size $m_2 \times m_2$. Finally, the resulting matrices are flattened and appended into an $(m_1^2 + m_2^2) \times N$ matrix, and the result is passed through a nonlinearity (ReLU).

As the input tensor is cyclically permuted according to a regular representation $\pi$ of $Z_{m_1 m_2}$, the output of this equivariant convolutional layer permutes according to the representation $\pi^{(1)} \oplus \pi^{(2)}$ where $\pi^{(1)}$ is the regular representation of $Z_{m_1}$ and $\pi^{(2)}$ is the regular representation of $Z_{m_2}$.

The proof that the fraction of separable dichotomies is given by $f(P, 2N)$ follows the same proof as for the standard periodic convolutions in Appendix A.4.1. Instead of a direct sum $\bigoplus_{k=1}^{N} \pi$ we get a direct sum $\bigoplus_{k=1}^{N} (\pi^{(1)} \oplus \pi^{(2)})$. Each of the $\pi^{(1)} \oplus \pi^{(2)}$ contain two trivial irreps in its decomposition, so that the final fraction is $f(P, 2N)$.

### A.4.5 INDUCED REPRESENTATIONS

First we state the definition of an induced representation. Let $H$ be a subgroup of a finite group $G$ and let $\rho : H \to \mathrm{GL}(W)$ be a representation of $H$. Let $V$ be the vector space of functions $f : G \to W$ such that $f(gh) = \rho(h)f(g)$ for all $h \in H$ and $g \in G$. We now define the induced representation $\pi : G \to \mathrm{GL}(V)$ to be the representation which satisfies $(\pi(g')f)(g) = f(gg')$.

For intuition, note that every element of $g$ can be written $g = rh$ where $r$ is a representative for a coset in $G/H$ and $h \in H$. This is because the cosets $G/H$ partition $G$ and the action of $H$ stays within a coset; hence $r$ selects out the coset, and $h$ goes to the desired element of the coset: $g = rh$. With this decomposition, the action of $\pi$ is then $\pi(rh)f(g) = \rho(h)f(gr)$. Hence the $r$ component under $\pi$ has the effect of permuting to the new coset that $gr$ belongs in, and the $h$ component under $\pi$ then has the effect $\rho(h)$ on the resulting vector $f(gr)$ that we originally specified. This is the most natural way to get a representation of $G$ from a representation of $H$. In the case of finite groups, one can think of the $r$ component as permuting a set of isomorphic copies of $V$, each copy corresponding to a different coset.

To compute the capacity of induced representations, and so prove Proposition 1, we use Frobenius reciprocity of characters. Recall that the character $\theta$ of a representation $\pi : G \to \mathrm{GL}(V)$ is the map $\theta : G \to \mathbb{C}$ induced by the trace: $\theta(g) = \mathrm{Tr}(\pi(g))$. Now let $\theta$ be the character of $\rho : H \to \mathrm{GL}(V)$ and let $\theta^G$ be the character of the induced representation. Then $\langle \theta^G(g) \rangle_{g \in G} = \langle \theta(h) \rangle_{h \in H}$ by Frobenius reciprocity of characters (Mackey, 1970). The average of the character is the number of trivial representations contained in the decomposition of the representation (see Serre (2014)). Hence the capacity of the induced representation is equal to the capacity of $\rho$. The existence of extensions beyond finite groups is not clear to the authors, but we welcome information if such exists.

