# OpenReview forum: "Capacity of Group-invariant Linear Readouts from Equivariant Representations: How Many Objects can be Linearly Classified Under All Possible Views?"
_ICLR.cc/2022/Conference — ICLR 2022 Poster_

### Official Review · Reviewer_zVT9 · 2021-11-02

**Correctness:** 4
**Technical Novelty And Significance:** 3
**Empirical Novelty And Significance:** Not applicable
**Recommendation:** 6
**Confidence:** 3

**Main Review:**

*Strengths*

** Fundamental problem: The paper studies a fundamental property of equivariant representations. To the best of my knowledge, their results are novel. The result is also quite elegant.

** Good writing: The authors did a great job writing the paper. Specifically, the mathematical load is reduced by deferring details to the appendix, and many examples and illustrations are given. For this type of theoretical paper, the authors did a very good job.

*Weaknesses*

** Relevance to current research/challenges: I am not sure how this theoretical question is actually related to the current methods and challenges in geometric/equivariant deep learning, which uses non-linear models. I feel that section 5 is supposed to answer but I am not sure I am convinced. Another question: why is the separation of equivariant (rather than invariant) representations important? Most models use an invariant representation of the object, where (to the best of my understanding) the \pi-manifold is trivial.




**Summary Of The Paper:**

This is a mainly theoretical paper that studies the capacity of equivariant representations, which is, the ability to linearly classify objects under different transformations of a symmetry group of interest.
The main result in the paper is showing that the number of dichotomies (different binary classifications) is related to the dimension of the fixed subspace of the representations, rather than the dimension of the representation.
The authors discuss several examples and also propose several usages of this theory to deep learning of symmetric objects: analysis of pooling operations and induced representations


**Summary Of The Review:**

The paper targets a fundamental problem regarding the separability of equivariant representations in a pretty general setup. The theoretical results are interesting and new (as far as I know), but the relevance to current research challenges is not clear. If this issue is addressed in a revision, I would be willing to upgrade my score.

after rebuttal:
I appreciate the authors' thoughtful response. I will keep my score at 6 at this point.

---

> ### Author Response · Authors · 2021-11-18
> **Response to Reviewer zVT9 (1/2)**
>
> Thank you for the thought-provoking review. We address the weaknesses stated in your review below. Please note that in our resubmitted manuscript, the Theorems, Lemmas, and Corollaries have been renumbered to not include the section number. We will use the new numbering in our responses below.
>
> > Relevance to current research/challenges: I am not sure how this theoretical question is actually related to the current methods and challenges in geometric/equivariant deep learning, which uses non-linear models. I feel that section 5 is supposed to answer but I am not sure I am convinced. Another question: why is the separation of equivariant (rather than invariant) representations important? Most models use an invariant representation of the object, where (to the best of my understanding) the $\pi$-manifold is trivial.
>
> Our theory is applicable to nonlinear models. Note that Figure 2 in Section 5 plots the fraction of linearly separable dichotomies throughout the layers of convolutional neural networks with nonlinearities, and the experiments match theory well for both random and pretrained CNNs. Reviewer s9Dq credits us for taking on nonlinear operations such as max pooling which are more challenging to address in theories of equivariant neural networks. Perhaps the reviewer is referring to the technique of using linear perceptrons as "linear probes" as opposed to nonlinear probes to evaluate capacity? There are admittedly trade-offs to considering different probes that capture different geometries of representations, such as nonlinear probes that seek to find nonlinear separatrices, and in the extreme measuring mutual information between the representation and labels. However, we do not feel that any one of these methods has a fundamental advantage over any others, and linear probes are a reasonable first thing to generalize to equivariant representations. See "Understanding intermediate layers using linear classifier probes" by Guillaume Alain and Yoshua Bengio for a discussion of the merits of using linear probes. Note also that in the revision we have extended our results to VC dimension, which may be more familiar to people in the ICLR community, and described its connection with perceptron capacity. This can be found in Section 3.2 and is discussed more thoroughly in our response to Reviewer  s9Dq. Finally, the reviewer may be referring to the linear nature of the group representations, where group actions are represented by matrices applied to vectors. Extensions to nonlinear group representations would be an interesting topic for future work, but we do not feel that this is essential for the results to be relevant to current research challenges since to the best of our knowledge linear group representation theory captures the structure of GCNNs currently under consideration by the community.
>
> Many models employ equivariant representations, as it is, for instance, important to preserve spatial information (at least up to a certain point). The intermediate layers of a CNN are equivariant, not invariant. The reviewer makes a good point in that the final layer often enforces invariance via a global average pooling operation. Our theory (Lemma 1) shows that the capacity after global average pooling is the same as before, implying that there is no loss of capacity. This helps make precise why this operation is a good thing to do, a point we now make more clearly in Section 5.3 (note that this section has been moved to Section 5.1.3. See our response to Reviewer bab9 for a more thorough description of this change). Note that as far as we are aware it may not always be immediately clear how best to enforce invariance in the final layer of a GCNN -- our theory shows that averaging over the group will always preserve perceptron capacity. Note that a final global max pooling layer, as opposed to global average pooling, would also enforce invariance, but would typically result in a reduced capacity, a point which is also made in the new Section 5.1.3.
>
> However, the capacity of the final layer is not the only relevant thing in a CNN. For instance, excessively tight bottlenecks in any layer can obfuscate or even destroy useful information. We believe that our theory is a valuable step toward quantifying what constitutes "excessively tight" while taking into consideration the equivariant structure of the representations. See our response to Reviewer s9Dq for a more thorough discussion of this point.
>
> Furthermore, our theory gives a useful notion of what it means for a CNN to be over/under parameterized (in our framework it's the number of filters that matters rather than the number of trainable parameters), which may be particularly salient when applied to the final convolutional layer. There are further challenges to making this connection precise, a point we discuss more thoroughly in our response to Reviewer s9Dq. These challenges are interesting topics for future work.(continued in next comment)

---

> > ### Author Response · Authors · 2021-11-18
> > **Response to Reviewer zVT9 (2/2)**
> >
> > To make these points clearer, we have added the following sentence to the Discussion: "The measures of expressivity explored here could prove valuable for ensuring that models with general equivariance relations have high enough capacity to support the tasks being learned, either by using more filters or by using representations with intrinsically higher capacity."

---

### Official Review · Reviewer_s9Dq · 2021-11-02

**Correctness:** 4
**Technical Novelty And Significance:** 3
**Empirical Novelty And Significance:** 3
**Recommendation:** 6
**Confidence:** 3

**Main Review:**

### Strengths
- This paper contributes to an underexamind area, theoretical analysis of expressivity of equivariant neural networks.  In particular, there is a large gap between what we know for non-equivariant and equivariant models, and this papers helps to close that gap.
- The definition of perceptron capacity for invariant models is logical and useful.
- The result the authors obtain for the capacity of invariant models is simple and easy to compute.  This means it is easier to interpret and potentially useful.
- I appreciate that operations such as pooling, which do not fit as nicely into the theory of equiv. NNs, but are still important, were considered.
- The empirical evaluation is quite convincing, suggesting the theory is correct.
- The related works section is actually a very nice and comprehensive survey of what (limited) theoretical work has been done on equivariant models.

### Weakness
- While I think it is important to examine the model capacity of equivariant models under the assumption of symmetrically distributed data, I’m not sure if perceptron capacity is the most relevant measure.  In some sense, this is evidenced by how simple the answer turns out to be depending only on the representation type of the final layer.  I wonder if other measures of capacity might be more relevant for understanding difference in model architecture?
- I’m not sure how much section 4 adds.  The examples are essentially just computing the multiplicity of the trivial rep in some group representations.  This is fairly straightforward using the peter-weyl theorem or characters.

### Questions
- What are the practical implications of the main theorem?   Is it better to add extra copies of the trivial representation to the final layer of the NN?
- Would the analysis of local pooling also be relevant to strided convolutions?
- If we assume the perceptron given by w is invariant, does this affect the result? (I guess not.)


### Small Notes
- Some of the citations in section 3 are really to references, not to the authors of the results.  This is okay, but it should be clarified.
- the definition of fixed point subspace does not need “largest”.  The definition is pointwise, so you can just define $W = \lbrace w \in V | g w = w \text{ for all } g \in G\rbrace$.
- Eqn 5 has no quantifier on h
- I don’t understand the argument for the upper bound in 5.1.2.  In particular, i’m not sure what is meant by “limiting case” or the set of points is that are equal.  Can you clarify it for me?
- 5.2: Should it be $\rho : H \to GL(V)$ since it acts on V?
- 5.2: It might be good to state a formal proposition here.




**Summary Of The Paper:**

This paper addresses an underexamined area, the question of expressivity in equivariant neural networks.  In general, there is a large gap between our theoretical understanding of expressivity and generalization for non-equivariant networks and that for equivariant ones; this paper helps to close the gap.  In particular, given a set of equivariant representations coming from an equivariant architecture, the authors determine how expressive these features are for an invariant binary classification problem.  To do so, they define a reasonable generalization of perceptron capacity for invariant classifiers and compute it.  The answer turns out to be relatively simple and is mainly determined by the multiplicity of the trivial representation in the group representation type of the equivariant representation, which is not difficult to compute.  The authors also consider the case of useful but not quite equivariant operations such as pooling. Lastly, they provide a convincing empirical verification of their results.

**Summary Of The Review:**

I tend towards accept due to the importance of making progress on what I see an underserved area.  However, I have some questions about the practical consequences of the work and whether other definitions of expressivity might have more potential.

---

> ### Author Response · Authors · 2021-11-18
> **Response to Reviewer s9Dq (1/4)**
>
> Thank you for the thoughtful review. We respond to each of the listed weaknesses below. Please note that in our resubmitted manuscript, the Theorems, Lemmas, and Corollaries have been renumbered to not include the section number. We will use the new numbering in our responses below.
>
> > While I think it is important to examine the model capacity of equivariant models under the assumption of symmetrically distributed data, I’m not sure if perceptron capacity is the most relevant measure. In some sense, this is evidenced by how simple the answer turns out to be depending only on the representation type of the final layer. I wonder if other measures of capacity might be more relevant for understanding difference in model architecture?
>
> The primary other measures of capacity that we are familiar with are VC/shattering dimension and related notions.
> Assuming the reviewer was referring primarily to this, we have retitled Section 3.2 from "Relationship with Cover's Theorem" to "Relationship with Cover's Theorem and VC dimension", reworked the section, and now include a discussion comparing the VC dimension of perceptrons with perceptron capacity. In summary, perceptron capacity is a strictly more informative measure of expressivity compared to VC dimension and behaves more sensibly in the thermodynamic limit of $N,P\to \infty$ with $P/N=O(1)$.
> The added text reads:
> "The VC dimension (Vapnik & Chervonenkis, 1968) of the perceptron trained on points in dimension $N$ is defined to be the largest number of points $P$ such that all dichotomies are linearly separable for some choice of the $P$ points. These points can be taken to be in general position.\footnote{This is because linear dependencies decrease the number of separable dichotomies (see Hertz et al., 2018).} Because $f(P,N)=1$ precisely when $P \leq N$, the VC dimension is always $N=P$ for any finite $N$ and $P$ (see Abu-Mostafa et al., 2012)), even while the asymptotics of $f$ reveal that $f(P,N)$ approaches $1$ at any $N>P/2$ as $N$ becomes large. In this sense the perceptron capacity yields strictly more information than the VC dimension, and becomes comparatively more descriptive of the expressivity as $N$ becomes large."
>
> In addition, we provide a definition of VC dimension for the $G$ invariant binary classification problem and have included a new Corollary (Corallary 1 in the updated manuscript) which states that the VC dimension of group-invariant perceptrons is $P = N_0$.
>
> You may also be interested in our response to Reviewer zVT9 where we briefly discuss measuring capacity with nonlinear probes, as opposed to the linear ones used here. If the reviewer is aware of other promising notions of capacity we would be grateful to hear about them so that they can be discussed in the Discussion.
>
>
> Our results are simple in the sense that they are clean and relatively straightforward to prove with group representation theory. We hope that this is evidence of the results being natural and fundamental and don't feel that it implies that they are not relevant. One complaint about relevance that is frequently voiced is that naive considerations of VC dimension and perceptron capacity don't take into account structure in the data. Here in contrast we explicitly take into account equivariant structure (i.e. labels are not distributed randomly, but are consistent within $\pi$-manifolds). Going beyond this to take into account even more structure is an interesting topic for future work.
>
> Our results are applicable to many different representations, including all intermediate layers of a deep neural network (indeed Figure 2 illustrates the behavior in successively deeper layers and indicates that max pooling has the effect of reducing capacity). While the relevance of linear probes is easy to see for final layers since the final layer of the network is a linear readout, they are also relevant for shallower layers. This point is discussed more thoroughly in the paper "Understanding intermediate layers using linear classifier probes" by Guillaume Alain and Yoshua Bengio, which we cite in our paper. (continued in next comment)

---

> > ### Author Response · Authors · 2021-11-18
> > **Response to Reviewer s9Dq (2/4)**
> >
> > For instance, one could speculate that if the capacity of an intermediate layer is excessively low, this can indicate an increased challenge for subsequent layers in learning the label structure in the representation (since a highly nonlinear function is required to do so). Put simply, excessively tight bottlenecks can obfuscate or even throw away useful information. Our theory is a useful step toward quantifying what constitutes excessively tight in the context of equivariant representations. Admittedly, more steps are needed to make such connections with trainability of deep neural networks more precise. For instance, as the reviewer alludes to, our theory does not fully connect the limitations to expressivity encountered in an intermediate bottleneck layer with the capacity of the final convolutional layer. We do not believe this limitation to be fundamental; appropriate modification and extension of the theory could bring us closer to a deeper understanding of such matters. This would be an interesting topic for future work.
> >
> > To reinforce our decision to study perceptron capacity and VC dimension and the utility our results have for the community, we have added the following sentence to the Discussion: "The measures of expressivity explored here could prove valuable for ensuring that models with general equivariance relations have high enough capacity to support the tasks being learned, either by using more filters or by using representations with intrinsically higher capacity."
> >
> > > What are the practical implications of the main theorem? Is it better to add extra copies of the trivial representation to the final layer of the NN?
> >
> > Adding extra copies of the trivial representation to the final layer of the NN (i.e. adding more filters) would increase the capacity of this final layer. Our theory takes steps toward quantifying how many filters are needed for the final linear readout layer to find separating hyperplanes. In some cases increasing the number of filters is good, i.e. pushing into the highly overparameterized regime that is thought to be beneficial in many cases; however, larger networks take more compute resources to train. Note that our theory suggests that the number of filters in a CNN is a more meaningful measure of overparameterization than simply the number of trainable parameters. Further analysis (such as taking into account multiple labels instead of dichotomies) is needed to make more concrete connections to over/under parameterization, which we leave to future work. See our response to the previous comment to see how we have modified the manuscript to make more explicit the practical implications of our theory.
> >
> > > Would the analysis of local pooling also be relevant to strided convolutions?
> >
> > Yes, this is a good observation. Corollary 2, which extends our results to subgroups, actually covers strided convolutions. We have added the sentence "For example, strided convolutions are equivariant to subgroups of the regular representation." before Corollary 2.
> >
> > > If we assume the perceptron given by w is invariant, does this affect the result? (I guess not.)
> >
> > We are in fact assuming that the output of the perceptron is invariant (a successful perceptron solution w should have the same readout for every point in a $\pi$-manifold, since every point in the manifold has the same label). Please let us know if we misunderstood your question.
> >
> > > Some of the citations in section 3 are really to references, not to the authors of the results. This is okay, but it should be clarified.
> >
> > This is a good point. We have now clarified that these are reference materials instead of the original works by saying "see [reference]" instead of just "[reference]".
> >
> > Edit: "as the author alludes to" -> "as the reviewer alludes to"

---

> > > ### Author Response · Authors · 2021-11-18
> > > **Response to Reviewer s9Dq (3/4)**
> > >
> > > > I’m not sure how much section 4 adds. The examples are essentially just computing the multiplicity of the trivial rep in some group representations. This is fairly straightforward using the peter-weyl theorem or characters.
> > >
> > > We agree that the answers to the examples we give are easy to compute using Theorem 1. However we still believe that these examples are essential for providing intuition to those newer to group representation theory. In addition, to our knowledge much of the current design of practical G-convolutional neural networks has been confined to the use of regular representations, a bias that we feel we need to account for by having these examples. The reality is that there is a very rich set of possible group representations to consider which give rise to less well-known G-CNN layer structure, such as the direct sum representation we highlight in Section 4.3. We give some justification for the merit of considering these less commonly considered equivariance relations by showing that they can be more expressive. To reinforce the relevance of this point, we have added the following sentence to Section 4.3: "These representations are used to build and test a novel G-convolutional layer architecture with higher capacity than standard CNN layers in Section 5." In addition, we point to connections with grid-cells in mammalian brains which have been shown to have certain positive qualities relative to place-cell codes (place-cell codes are analogous to regular representations of groups) and which are an area of high interest in the neuroscience community. We have also modified the last sentence of 4.3 to more clearly make this point: "These representations are analogous to certain formulations of grid cell representations as found in entorhinal cortex of rats (Hafting, 2005), which can have desirable qualities in comparison to place cell representations (Sreenivasan, 2011). Place cells are analogous to standard convolutional layers."
> > >
> > > Finally, we remark that each of the other official reviewers listed the examples in Section 4.3 as a strength of the paper. As such we feel that we should keep this section intact.

---

> > > > ### Author Response · Authors · 2021-11-18
> > > > **Response to Reviewer s9Dq (4/4)**
> > > >
> > > > > the definition of fixed point subspace does not need “largest”. The definition is pointwise, so you can just define $W=\\{w \in V | gw=w \text{ for all } g \in G\\}$.
> > > >
> > > > We changed the definition per your advice. It now reads:
> > > > "The **fixed point subspace** is the subspace $W=\\{w \in V | gw=w \text{ for all } g \in G\\}$."
> > > >
> > > > > Eqn 5 has no quantifier on h
> > > >
> > > > Fixed, thanks for catching this. We had just neglected to add $\forall h\in H$ to the statement.
> > > >
> > > > > I don’t understand the argument for the upper bound in 5.1.2. In particular, i’m not sure what is meant by “limiting case” or the set of points is that are equal. Can you clarify it for me?
> > > >
> > > > The lower bound in 5.1.2 considers the capacity of local nonlinear pooling. Unlike local/global average pooling which preserves the dimension of the fixed point subspace for the group $G$, nonlinear pooling operations do not in general. Section 5.1.2 considers a lower bound on the number of separable dichotomies using only the fact that the nonlinear pooled code is a representation of a subgroup $H$ of $G$. Since we maintain equivariance to $H$ after nonlinear pooling we can reuse our argument from the original case where $H=G$, but now averaging over $H$ rather than $G$.  Concretely, separability of the full orbits is equivalent to separability of the points
> > > > $$\\langle \\rho(h ) \\rangle_h  \\mathcal{P}( \\pi(j) r\^{\\mu} ) \\ , \ \forall j \in G/H \ ,  \forall \mu \in [P]$$
> > > > Intuitively, upon the addition of a single new manifold $P \to P+1$, there are (at most) $k=|G/H|$ additional constraints to this separability problem, one for each representative $\pi(j) r^{P+1}$ (in the original case when $G=H$ each new manifold adds a single additional constraint). We show in Appendix A.4.3 that this implies $f \geq  f_{cover}(P, N/k)$.
> > > >
> > > > The sentence about the limiting case refers to the *upper bound* on the fraction of separable dichotomies.  Here we show that the fraction of separable dichotomies $f$ cannot be improved by local nonlinear pooling: $f_{pooled} \leq f_{cover}(P,  N)$. This follows from the fact that separability over all $j$ of $\langle \rho(h ) \rangle_h  \mathcal{P}( \pi(j) r^\mu )$ implies separability of $\langle \rho(h ) \rangle_h  \mathcal{P}( r^\mu )$, and that the $H$-invariant problem (separating $\langle \rho(h ) \rangle_h  \mathcal{P}( r^\mu )$) has a fraction $f_{Cover}(P,N)$ of separable dichotomies. This bound can be made tight if all points of the form $\langle \rho(h ) \rangle_h  \mathcal{P}( \pi(j) r^\mu )$ are exactly equal for all $j \in G/H$ (which happens in average pooling). In this case, only 1 constraint is added upon the addition of a new manifold. Thus, this bound is exactly achieved for average pooling where $f_{pooled} =f_{cover}(P,N)$.
> > > >     To make these points clearer, in section 5.1.2, we replaced the old explanation with the following sentence "Alternatively, an upper bound $f \leq f(P,N)$ persists since a $w$ which satisfies 4 must separate $\langle\rho(h) \rangle_{h\in H} \mathcal{P}( r^\mu)$. This upper bound is tight when all $k$ points $\langle \rho(h) \rangle_{h\in H} \mathcal{P}(\pi(j)  r^\mu)$ coincide for each $\mu$, giving capacity $f(P,N)$.
> > > > This is what occurs in the average pooling case where the upper bound $f \leq f(P,N)$ is tight."
> > > >
> > > > > 5.2: Should it be $\rho : H \to GL(V)$ since it acts on V? It might be good to state a formal proposition here.
> > > >
> > > > $\rho(h)$ actually acts on $W$; the induced representation $\pi(g)$ acts on $V$. In the case where $G$ is finite, $V$ is isomorphic to $|G|/|H|$ copies of $W$. We agree that this is an easy misinterpretation to make. The induced representation is rather subtle and complex and probably outside the scope of the main text to present the idea well enough to a newcomer. Considering this we have moved the definition of induced representations to the Appendix (Appendix A.4.5) alongside the further discussion and intuition we give. In addition, we took your advice and now state a formal proposition of the result.

---

> > > > > ### Comment · Reviewer_s9Dq · 2021-11-22
> > > > > **induced reps**
> > > > >
> > > > > Quickly on the last point.  I believe the issue is still there; it has just moved to the appendix.   You write $f(gh) = \rho(h) f(g)$, but also $f(g) \in V$ and $\rho(h) \in GL(W)$.   I suppose the issue is that
> > > > >
> > > > > > Let $V$ be the linear space of functions $f : G \to V$ ...
> > > > >
> > > > > should be  Let $V$ be the linear space of functions $f : G \to W$ ...
> > > > >
> > > > > (Also, given that induced representations are at the heart of steerable CNN, I wouldn't necessarily consider them so exotic.)

---

> > > > > > ### Author Response · Authors · 2021-11-22
> > > > > > **response to "induced reps"**
> > > > > >
> > > > > > Thank you for being persistent with this! You are correct: the function should be $f: G \to W$. This has been fixed in the most recent update to the revision. Please let us know if anything else looks amiss in the presentation of the induced representation.
> > > > > >
> > > > > > It is a good point that induced representations are core to the theory of steerable CNNs. Do you recommend placing the definition back in the main text?

---

> > > > > > > ### Comment · Reviewer_s9Dq · 2021-11-28
> > > > > > > **Thanks**
> > > > > > >
> > > > > > > > It is a good point that induced representations are core to the theory of steerable CNNs. Do you recommend placing the definition back in the main text?
> > > > > > >
> > > > > > > I think the current version is good.  It's a good case to consider and I like the proposition.
> > > > > > >
> > > > > > > > We are in fact assuming that the output of the perceptron is invariant (a successful perceptron solution w should have the same readout for every point in a -manifold, since every point in the manifold has the same label). Please let us know if we misunderstood your question.
> > > > > > >
> > > > > > > I mean considering only $w$ in the definition such that $\tilde w$ = $w$ (invariant as $w = w\pi(g)$).   Since an equivariant method would likely have such a constraint in its last layer, I wondered if it might make sense to impose it in the definition.  I guessed though that it wouldn't make a difference to the answer though since the dichotomies are actually realized by such invariant $w$.
> > > > > > >
> > > > > > > ### Thanks for edits and answers
> > > > > > >
> > > > > > > Thank you the authors for the answers, clarifications, and edits to the paper.    I particularly appreciate the discussion of VC dimension, the discussion of applying the method to intermediate layers and the questions related to bottlenecks, and the clarification of 5.1.2.   I have read all of the reviews and responses.  All in all, I feel the work is solid and should be accepted.

---

> > > > > > > > ### Author Response · Authors · 2021-12-01
> > > > > > > > **Invariant perceptron**
> > > > > > > >
> > > > > > > > Yes, so our constraint is that $y^{\mu}w^T\pi(g)x^{\mu}>0$ (the output of the perceptron is invariant) and if I understand correctly your point is that we could have perhaps instead imposed the stronger constraint that the output weights satisfy $w^T = w^T\pi(g)$. As you say, combining the linear operation of global average pooling and a linear readout, as in $\tilde{w}=\langle \pi \rangle^{T}w$, is similar to having the constraint $\tilde{w}^{\top} = \tilde{w}^{\top}\pi(g)$, and this is a common way to achieve invariance. Our problem formulation ($y^{\mu}w^T\pi(g)x^{\mu}>0$) is more general than this and contains the stronger constraint (($w^T = w^T\pi(g)) \land (y^{\mu}w^Tx^{\mu}>0$)) as a special case. In addition, this special case is constructed explicitly in the proof of Lemma 1. Hopefully this connection is now clearer in the new Section 5.1.3. Please let us know if you feel like this connection is still not clear after the addition of Section 5.1.3.

---

### Official Review · Reviewer_bab9 · 2021-11-02

**Correctness:** 4
**Technical Novelty And Significance:** 3
**Empirical Novelty And Significance:** 3
**Recommendation:** 8
**Confidence:** 3

**Main Review:**

Strengths:
- The paper is well embedded in literature and the problem formulation is clear.
- The presentation is clear (though some jargon could be better explained, see below).
- The examples are helpful.
- The paper is reproducible (see attached code).
- The paoer us well written and intuitive.

Weakness:
- The presentation of the main idea could be better organized (see minor comments below).
- The result is quite intuitive in hindsight, but when reading the paper I struggled with notions of averaging representations over G. I started working out what it meant and figured that for non-trivial irreps the average is zero (for compact groups at least). Then, only later on in the paper (page 6) it is indeed mentioned that his is the case. This kind of intuition would have been nice to receive early on since, if I understood correctly, this is at the core of the approach: it explains why it is all about the trivial irreps (all about the invariant subspaces).
- I'm not fully sure about the relevance of the "one-shot learning" section 5.3. It seems to avoid the need to generate (augment?) the entire dataset/orbits from the reference points $r^\mu$. However, since the capacity is determined by invariant subspaces, and one typically indeed designs G-CNNs that are at the final layer invariant, each point in the orbit will be mapped to the same representation anyway. Does the algorithm therefore solve an non-existing problem? (I'm probably misinterpreting here and would be happy to stand corrected if so)

Regarding Jargon:
It would be nice to help readers that are new to these kind of analysis with a bit more intuition on some terminology.
- E.g., it would have been nice to provide a definition of “dichotomy”, it took me a while to figure out what is precisely meant with it (still not fully sure). Is it just the dataset, pairs of equivalence classes + label. Or is it an instance of a possible dataset?
- Hopefully the “fraction of linearly separable dichotomies” (top page 5) is then also clearer.
- What is “a datapoint in general position”? (general position?)
- Section 4.3. What does coprime mean?

Minor points:
- Section 4.3. What is idea behind the direct sum representations, isn’t this just considering the direct product of two groups that may as well be analyzed independently.
- Typo last sentence 5.1.1. “dichotomies if” -> “dichotomies is”
- I find Figure 2 hard to interpret, mainly because of the vertical axis being “capacity”. When saying things like “max pooling reduces capacity by a factor between ¼ and 1” I expect this factor to be apparent on the vertical axis, but it seems to refer to the bending point (at some alpha) where capacity dives below 0.5. I think the further this point is to the right, the more complex datasets the method could represent and it indeed reflects capacity in that sense. But then the vertical axis is confusing to me, shouldn’t this be the fraction f?
- One of the conclusions is that pooling may reduce capacity. The subsequent sentence says “This quantification of expressivity could prove a valuable tool in building new G-CNNs…” In what sense is it valuable. Could you be more explicit about what possible implications does this have on NN design.





**Summary Of The Paper:**

The paper provides an analysis of network capacity of group equivariant NNs. The analysis is based on determining the fraction of linearly separable dichotomies. It is found that this fraction is determined by the the number of trivial irreps that appear in the decomposition of the representations into irreps. I.e., the fraction is determined by the number of sub-spaces that are left-invariant by the group action. The paper is intuitive, well written, and theoretical results are confirmed by experiments.

**Summary Of The Review:**

All in all the paper is a nice read and provides interesting insights. The paper is sound and the proofs are good to follow. I consider it a high quality submission that could, however, still be improved a bit in its presentation.

---

> ### Author Response · Authors · 2021-11-18
> **Response to Reviewer bab9 (1/3)**
>
> Thank you for the encouraging and constructive review. We address the listed weaknesses in order below. Please note that in our resubmitted manuscript, the Theorems, Lemmas, and Corollaries have been renumbered to not include the section number. We will use the new numbering in our responses below.
>
> > The presentation of the main idea could be better organized (see minor comments below).
>
> We respond to each of these comments in turn below.
>
> > The result is quite intuitive in hindsight, but when reading the paper I struggled with notions of averaging representations over G. I started working out what it meant and figured that for non-trivial irreps the average is zero (for compact groups at least). Then, only later on in the paper (page 6) it is indeed mentioned that his is the case. This kind of intuition would have been nice to receive early on since, if I understood correctly, this is at the core of the approach: it explains why it is all about the trivial irreps (all about the invariant subspaces).
>
> This is an astute observation. We have taken your advice to improve upon the presentation of the main idea. We note that the point about nontrivial irreps averaging to zero is actually first described on page 5 in the proof of Theorem 1, and that this discussion follows almost immediately after irreps are first introduced. Hence we feel that this is the earliest that this point could reasonably be made. However, we realize that it is not reasonable to expect a reader to absorb a main idea if it is embedded in a proof, so we now reiterate the importance of this point immediately following the proof in a new Remark:
> "**Remark**: A main idea used in the proof is that only nontrivial irreps average to zero. This will be illustrated with examples in Section 4 below."
>
> > E.g., it would have been nice to provide a definition of “dichotomy”, it took me a while to figure out what is precisely meant with it (still not fully sure). Is it just the dataset, pairs of equivalence classes + label. Or is it an instance of a possible dataset?
>
> We have now included a definition of dichotomy in the introduction:
> "Our starting point is the classical notion of the **perceptron capacity** (sometimes also known as the fractional memory/storage capacity) -- a quantity fundamental to the task of object categorization. Defined as the maximum number of points for which all (or 1-$\delta$ fraction of) possible binary label assignments (i.e. **dichotomies**) afford a hyperplane that separates points with one label from the points with the other, it can be seen to offer a quantification of the expressivity of a representation."
> In addition, we have added a Glossary (Appendix 1) which has the definition of dichotomy stated there: "Dichotomy: a particular binary labeling $\{ y^\mu \}$ of $P$ points $\{x^\mu\}$."
>
> > Hopefully the “fraction of linearly separable dichotomies” (top page 5) is then also clearer.
>
> The inclusion of our definition hopefully makes the statement of what a dichotomy is in Section 2 more clear:
> "The condition for realizing the dichotomy $\\{y^{\mu}\\}\_\{\mu\}$ ..."
> Here the dichotomy is the "binary label assignment" $\\{y^{\mu}\\}\_\{\mu\}$. So a dichotomy is defined in reference to an already defined dataset, where it is a particular partitioning of this dataset into two parts (e.g. the binary labeling). We hope that this eliminates the confusion surrounding this terminology. Please let us know if this is still not the case.
>
> > What is “a datapoint in general position”? (general position?)
>
> We have added a definition of general position as a footnote to the first sentence of section 3.2, along with a short statement about how unconstraining this condition is:  "A set of $P$ points is in **general position** in $N$-space if every subset of $N$ or fewer points is linearly independent. This says that the points are "generic" in the sense that there aren't any prescribed special linear relationships between them beyond lying in an $N$-dimensional space. Points drawn from a Gaussian distribution with full-rank covariance are in general position with probability one." This definition also appears in a new Glossary (Appendix 1). Please see our response to Reviewer hH77 for additional discussion as to why this is as weak of a condition as can reasonably be assumed without imposing structure in the data beyond the equivariance condition.

---

> > ### Author Response · Authors · 2021-11-18
> > **Response to Reviewer bab9 (2/3)**
> >
> > > I'm not fully sure about the relevance of the "one-shot learning" section 5.3. It seems to avoid the need to generate (augment?) the entire dataset/orbits from the reference points. However, since the capacity is determined by invariant subspaces, and one typically indeed designs G-CNNs that are at the final layer invariant, each point in the orbit will be mapped to the same representation anyway. Does the algorithm therefore solve an non-existing problem? (I'm probably misinterpreting here and would be happy to stand corrected if so)
> >
> > Thank you for stimulating further consideration of Section 5.3, especially in light of the insightful connections you draw with global average pooling that is now standard in CNNs (note that as far as we are aware this was not necessarily the case for, say, VGG, and justification in the literature for why this operation is "okay" is generally intuitive and not very precise). Our theory indeed shows that classifying the $\pi$-manifolds (i.e. data "augmented" to be at every possible position specified by the equivariance relationship) is in a precise sense equivalent to classifying the representation after global average (over the group) pooling. The first step in our algorithm is in fact to do a global average pooling.
> >
> > We think the connections you highlighted are important to point out, so we have (1) changed the title of Section 5.3 from "One-shot learning of readout weights" to "Global pooling", (2) moved the section to 5.1.3 to be adjacent to the other sections about pooling, (3) removed the explicit enumerated algorithm from the section, and (4) reworked the section to read like this:
> > "Lemma 1 shows that linear separability of $\pi$-manifolds is equivalent to linear separability after global average pooling (i.e. averaging the representation over the group). This is relevant to CNNs, which typically perform global average pooling following the final convolutional layer. This strategy is efficient: Lemma 1 implies that it allows the learning of optimal linear readout weights following the pooling operation, given only a single point from each $\pi$-manifold (i.e. no need for data augmentation). Global average pooling reduces the problem of linearly classifying $|G|P$ points in a space of dimension $N$ to that of linearly classifying $P$ points in a space of dimension $\dim V_0$. Note that by an argument similar to Section 5.1.2, global max pooling may in contrast reduce capacity."
> >
> > Hence we give some formal justification for why this speedup operation is okay to do in deep neural networks. Note that other operations that enforce invariance, such as global max pooling, would not necessarily preserve capacity. Section 5.1.3 formalizes how to extend global average pooling to general GCNNs, and in particular to the situation where the trivial irreps are not fully aligned with the output channels of the G-convolution.
> >
> > > Section 4.3. What does coprime mean?
> >
> > We have added a definition of coprime: "Two numbers are coprime if they have no common prime factor."
> >
> > **Minor points:**
> > > Section 4.3. What is idea behind the direct sum representations, isn’t this just considering the direct product of two groups that may as well be analyzed independently.
> >
> > The idea behind the direct sum representation is that it is an example of a group representation formed by permutation matrices (like the regular representation), but that has a different capacity than the regular representation. Because it can be encoded by permutation matrices, equivariant representations can be implemented by a sliding window convolutional neural network layer (including element-wise nonlinearities) with some additional structure. The main purpose of this example is to show that for G-convolutions of potential practical interest, the capacity is not always given purely in terms of the number of output channels. This allows for the building of, for instance, G-convolutions that have a higher capacity than the number of output channels. To help make this point clearer, we have added the following sentence to Section 4.3: "These representations are used to build and test a novel G-convolutional layer architecture with higher capacity than standard CNN layers in Section 5."
> >
> > We also think that this section is interesting to readers with a neuroscience background, as these representations are analogous to certain formulations of grid cell representations, which have been shown to have many desirable properties in comparison to place cell representations (place cell representations are analogous to standard CNNs). To make this point clearer, we have reworked this description in the manuscript in Section 4.3 to read "These representations are analogous to certain formulations of grid cell representations as found in entorhinal cortex of rats (Hafting, 2005), which can have desirable qualities in comparison to place cell representations (Sreenivasan, 2011). Place cells are analogous to standard convolutional layers."

---

> > > ### Author Response · Authors · 2021-11-18
> > > **Response to Reviewer bab9 (3/3)**
> > >
> > > > Typo last sentence 5.1.1. “dichotomies if” -> “dichotomies is”
> > >
> > > Fixed, thanks for catching this.
> > >
> > > > I find Figure 2 hard to interpret, mainly because of the vertical axis being “capacity”. When saying things like “max pooling reduces capacity by a factor between ¼ and 1” I expect this factor to be apparent on the vertical axis, but it seems to refer to the bending point (at some alpha) where capacity dives below 0.5. I think the further this point is to the right, the more complex datasets the method could represent and it indeed reflects capacity in that sense. But then the vertical axis is confusing to me, shouldn’t this be the fraction f?
> > >
> > > The labeling of the y axis in Figure 2 was actually quite problematic, and we thank the reviewer for pointing this out (we noticed this as well, but only after the submission deadline). This y axis is indeed the fraction of separable dichotomies, so we have relabeled it $f(\\alpha)$. The capacity is determined by the value of alpha at which $f(\\alpha)=1-\\delta$, where $\\delta$ is some specified small value. In the limit as P and N go to infinity while $P/N =O(1)$, the precise value of $\\delta$ no longer matters since $f(\\alpha)$ becomes a step function. The code for labeling the plots in the supplementary material has also been fixed to reflect this.
> > >
> > > > One of the conclusions is that pooling may reduce capacity. The subsequent sentence says “This quantification of expressivity could prove a valuable tool in building new G-CNNs…” In what sense is it valuable. Could you be more explicit about what possible implications does this have on NN design.
> > >
> > > Capacity is closely related to the ability of a network to "memorize" a dataset -- in other words, it is one way to determine if a network is over/under parameterized that is naive to structure between inputs and class labels. Note that the classical perceptron capacity is closely related to the VC/shattering dimension of the perceptron (see our response to Reviewer hH77 for a discussion of this point and a description for how we have featured VC dimension more prominently in the revision). To make this point clearer, we have added the following sentence to the Discussion: "The measures of expressivity explored here could prove valuable for ensuring that models with general equivariance relations have high enough capacity to support the tasks being learned, either by using more filters or by using representations with intrinsically higher capacity."
> > > When building new GCNNs, one will want to ensure that the capacity is high enough to support the task, either by using more filters or by using representations with intrinsically higher capacity (such as the direct sum representations we illustrate in Section 4.3). Our theory is a step in the direction of being able to calibrate the expressivity of a GCNN to fit the needs of the task. Our theory also clarifies why it makes sense to do global average pooling operations after the final convolutional layer in CNNs, prescribes the proper way to do global average pooling in GCNNs with more complex equivariance relations (average over the group), and points out that global max pooling could in contrast reduce complexity. Please see our response above for the details of how we have modified the main text to better communicate these points about global pooling.

---

> > > > ### Comment · Reviewer_bab9 · 2021-11-19
> > > > **Thank you for addressing my points and for the paper improvements**
> > > >
> > > > I very much appreciate the effort the authors put in answering my questions and using the opportunity to improve the paper.
> > > >
> > > > In my opinion, the changes were effectively applied and *greatly improved the quality of the paper, which was content-wise already very good*.
> > > >
> > > > As my initial review was already postive and anticipated improvements in presentation, I keep my evaluation of the paper at an 8.

---

### Official Review · Reviewer_hH77 · 2021-11-03

**Correctness:** 3
**Technical Novelty And Significance:** 3
**Empirical Novelty And Significance:** 2
**Recommendation:** 6
**Confidence:** 3

**Main Review:**

**Strengths**:
- The authors give a fantastic lemma which implies that classifying the $\pi$-manifolds is equivalent to classifying their centroids.
- The authors give some useful examples to help to understand. They then also compare the expressivity of some different representations.
- The idea is novel. This paper gives a new perception of how to describe the expressivity of a representation.

**Weaknesses**:
- Some mentioned notations are not defined in the main text, even if they are in the main theorem. It brings some problems for understanding. I can not find the definitions of "in general position" and "range" in theorem 3.2. If there is a notation table, it may be easier for reading and understanding.
- Theorem 3.2 is true only when the centroids are in general position. Since I can not find the mathematical definition of "in general position", I do not know whether this assumption is too strong.
- The authors do not give further theoretical analysis about Eq. (1).
- The authors propose an algorithm to classify the $\pi$-manifolds and claim it is efficient. Although the algorithm seems efficient, the authors did not give any analysis about the time complexity.

**Summary Of The Paper:**

This paper studies the expressivity of a representation constrained by group equivariance. The authors first use a classical notion of the perceptron capacity to offer a quantification of the expressivity. Then, an algorithm is designed to efficiently classify the $\pi$-manifolds. Besides, the authors further apply the theoretical results to some practical examples and give bounds on the capacity when pooling operators exist.

**Summary Of The Review:**

Overall, I vote for rejection. The idea is novel and interesting. However, I think this work lack sufficient theoretical and empirical analyses.

-----

The authors have carefully addressed my concern. My score is thus raised to 6.

---

> ### Author Response · Authors · 2021-11-18
> **Response to Reviewer hH77 (1/2)**
>
> Thank you for the helpful comments. Please see below for our responses to each of the weaknesses listed. Also please note that in our resubmitted manuscript, the Theorems, Lemmas, and Corollaries have been renumbered to not include the section number. We will use the new numbering in our responses below.
>
> > Some mentioned notations are not defined in the main text, even if they are in the main theorem. It brings some problems for understanding. I can not find the definitions of "in general position" and "range" in theorem 3.2. If there is a notation table, it may be easier for reading and understanding.
>
> We have added a definition of general position as a footnote to the first sentence of Section 3.2, along with a short statement about how unconstraining this condition is:  "A set of $P$ points is in **general position** in $N$-space if every subset of $N$ or fewer vectors is linearly independent. This says that the points are "generic" in the sense that there aren't any prescribed special linear relationships between them beyond lying in an $N$-dimensional space. Points drawn from a Gaussian distribution with full-rank covariance are in general position with probability one."
> We have also added a definition for "range" inline in the statement for Theorem 1. The range of a matrix M that maps vector space V to vector space W is {$\{Mx : x \in V\}$}.
>
> We have added a glossary for definitions and notation in the Appendix (this is now Appendix 1).
>
>
> > Theorem 3.2 is true only when the centroids are in general position. Since I can not find the mathematical definition of "in general position", I do not know whether this assumption is too strong.
>
> See above for a definition of general position which is now included in the manuscript. This assumption is the loosest possible condition that can be expected without enforcing structure in the data beyond our equivariance condition. The condition of general position of a set of points means in essence that there are no prescribed linear dependence relations between the points (beyond the statement of what linear subspace these points belong to). Points drawn from full-dimensional Gaussian distributions are in general position with probability 1. Considering all this, we believe the general position condition to be very loose and not constraining of the generality of the results in any meaningful way.
>
> > The authors do not give further theoretical analysis about Eq. (1).
>
> We have included some additional analysis of Eq. (1) by highlighting the sigmoidal shape of the curve $f(P,N)$ for fixed $N$ and varying $P$, and pointing out connections with VC dimension. After introducing $f(P,N)$, Section 3.2 now reads
> "Taking $\alpha = P/N$, the function $f$ is a nonincreasing sigmoidal function of $\alpha$ that takes the value 1 when $\alpha \leq 1$, 1/2 when $\alpha=2$, and approaches $0$ as $\alpha \to \infty$. In the thermodynamic limit of $P, N \to \infty$ and $\alpha=O(1)$, $f(\alpha)$ converges to a step function (Gardner, 1987; 1988; Shcherbina, 2003) $\bar{f}(\alpha) = \lim_{P,N \to \infty, \alpha = P/N} f(P,N) =\Theta(2-\alpha)$ indicating that in this limit all dichotomies are realized if $\alpha < 2$ and no dichotomies can be realized if $\alpha>2$, making $\alpha_c = 2$ a critical point in this limit.
>
> The VC dimension (Vapnik & Chervonenkis, 1968) of the perceptron trained on points in dimension $N$ is defined to be the largest number of points $P$ such that all dichotomies are linearly separable for some choice of the $P$ points. These points can be taken to be in general position.\footnote{This is because linear dependencies decrease the number of separable dichotomies (see Hertz et al., 2018).} Because $f(P,N)=1$ precisely when $P \leq N$, the VC dimension is always $N=P$ for any finite $N$ and $P$ (see Abu-Mostafa et al., 2012)), even while the asymptotics of $f$ reveal that $f(P,N)$ approaches $1$ at any $N>P/2$ as $N$ becomes large. In this sense the perceptron capacity yields strictly more information than the VC dimension, and becomes comparatively more descriptive of the expressivity as $N$ becomes large."
>
> We feel that this is the appropriate amount of analysis considering the goals and setting of our paper. Please point us to additional ways of analyzing Equation (1) if you feel they would be relevant.

---

> > ### Author Response · Authors · 2021-11-18
> > **Response to Reviewer hH77 (2/2)**
> >
> > > The authors propose an algorithm to classify the $\pi$-manifolds and claim it is efficient. Although the algorithm seems efficient, the authors did not give any analysis about the time complexity.
> >
> > In response to this and comments from other reviewers, we have (1) changed the title of Section 5.3 from "One-shot learning of readout weights" to "Global pooling", (2) moved the section to 5.1.3 to be adjacent to the other sections about pooling, and (3) removed the explicit enumerated algorithm from the section.
> > To more clearly highlight connections with global average pooling, we have replaced our enumerated algorithm from before with an explicit description of our algorithm as the strategy of doing linear classification on data after first performing global average pooling. We have explicitly described the time complexity of this strategy in terms of the time complexity of doing a global average pooling followed by training a linear classifier, which we feel is the correct level of description for this setting. You may also be interested in reading our response to Reviewer bab9, where we describe connections of this section (now 5.1.3) with global average pooling (and max average pooling) in GCNNs. The modified section now reads like this:
> > "Lemma 1 shows that linear separability of $\pi$-manifolds is equivalent to linear separability after global average pooling (i.e. averaging the representation over the group). This is relevant to CNNs, which typically perform global average pooling following the final convolutional layer, which is typically justified by a desire to achieve translation invariance. This strategy is efficient: Lemma 1 implies that it allows the learning of optimal linear readout weights following the pooling operation, given only a single point from each $\pi$-manifold (i.e. no need for data augmentation). Global average pooling reduces the problem of linearly classifying $|G|P$ points in a space of dimension $N$ to that of linearly classifying $P$ points in a space of dimension $\dim V_0$. Note that by an argument similar to Section 5.1.2, global max pooling may in contrast reduce capacity."
> >
> > In general, we believe that we have addressed the specific weaknesses pointed out in the review. We hope that this is sufficient grounds for reevaluation of the statement that "...this work lack sufficient theoretical and empirical analyses." If there are other specific directions that the Reviewer thinks would add to the paper, we are open to hearing them.

---

> > > ### Comment · Reviewer_hH77 · 2021-11-25
> > > **Concerns addressed**
> > >
> > > Thanks for the detailed resposes. All my concerns have been addressed. My score will be raised to 6.

---

### Decision · Program_Chairs · 2022-01-20

**Decision:**

Accept (Poster)

**Comment:**

The authors' provide a discussion of Cover's Theorem in the setting of equivariance.  The reviewers consider the work well explained and interesting, especially after the revisions, and so I will vote to accept.